# OpenOmni: Advancing Open-Source Omnimodal Large Language Models with Progressive Multimodal Alignment and Real-time Emotional Speech Synthesis

**Run Luo**[1,2,3][*] **Ting-En Lin**[4], **Haonan Zhang**[4], **Yuchuan Wu**[4], **Xiong Liu**[4],
**Yongbin Li**[4][†], **Longze Chen**[1,2], **Jiaming Li**[1,2], **Lei Zhang**[1,2], **Xiaobo Xia**[5,6],
**Hamid Alinejad-Rokny**[7], **Fei Huang**[4], **Min Yang**[1,3][†]

[1]Shenzhen Key Laboratory for High Performance Data Mining,
Shenzhen Institute of Advanced Technology, Chinese Academy of Sciences
[2]University of Chinese Academy of Sciences
[3]Shenzhen University of Advanced Technology     [4]Tongyi Lab, Alibaba Group
[5]National University of Singapore     [6]MoE Key Laboratory of Brain-inspired Intelligent
Perception and Cognition, University of Science and Technology of China
[7]University of New South Wales, Sydney, New South Wales, Australia
{r.luo, min.yang}@siat.ac.cn   {ting-en.lte, shuide.lyb}@alibaba-inc.com

## Abstract

Recent advancements in omnimodal learning have significantly improved understanding and generation across images, text, and speech, yet these developments remain predominantly confined to proprietary models. The lack of high-quality omnimodal datasets and the challenges of real-time emotional speech synthesis have notably hindered progress in open-source research. To address these limitations, we introduce OpenOmni, a two-stage training framework that integrates omnimodal alignment and speech generation to develop a state-of-the-art omnimodal large language model. In the alignment phase, a pretrained speech model undergoes further training on image-text tasks, enabling (near) zero-shot generalization from vision to speech, outperforming models trained on tri-modal datasets. In the speech generation phase, a lightweight decoder is trained on speech tasks with direct preference optimization, which enables real-time emotional speech synthesis with high fidelity. Extensive experiments demonstrate that OpenOmni surpasses state-of-the-art models across omnimodal, vision-language, and speech-language benchmarks. It achieves a 4-point absolute improvement on OmniBench over the leading open-source model VITA, despite using 5× fewer training examples and a smaller model size (7B vs. 7×8B). Besides, OpenOmni achieves real-time speech generation with less than 1 second latency at non-autoregressive mode, reducing inference time by 5× compared to autoregressive methods, and improves emotion classification accuracy by 7.7%. The codebase is available at https://github.com/RainBowLuoCS/OpenOmni.

## 1 Introduction

The success of large language models (LLMs) [1, 2, 3, 4, 5, 6, 7, 8, 9, 10] has driven rapid advancements in multimodal large language models (MLLMs) [11, 12, 13, 14, 15, 16, 17], partic-

---

[*]Work done while interning at Tongyi Lab.

[†]Min Yang and Yongbin Li are corresponding authors.

39th Conference on Neural Information Processing Systems (NeurIPS 2025).

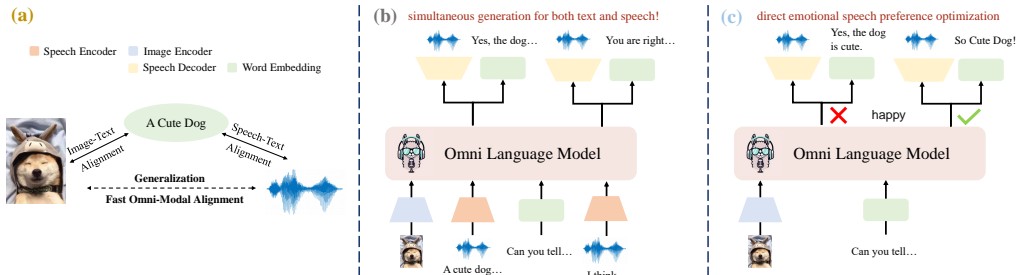

Figure 1: **Overview of the motivation and architecture of OpenOmni.** (a) OpenOmni adopts a progressive alignment strategy to generalize from vision-language to speech-language tasks, avoiding the need for costly tri-modal datasets and resources. (b) OpenOmni integrates a lightweight end-to-end speech decoder, enabling parallel text and speech generation while effectively reducing inference latency. (c) By utilizing DPO, OpenOmni generates emotionally coherent and context-aware speech without relying on additional control modules or handcrafted prompts. For simplicity, our core architecture is presented without the connectors between modules.

ularly in vision-language models (VLMs) [18, 19, 20, 21, 22, 23] and speech-language models (SLMs) [24, 25, 26, 27, 28]. These innovations mark a paradigm shift in machine understanding and human-computer interaction, fueling interest in omnimodal large language models (OLLMs), which are models that integrate vision, language, and speech into a unified system. The emergence of GPT-4o underscores the potential of holistic multimodal AI, yet open-source alternatives remain significantly behind.

Despite their promise, existing open-source OLLMs [29, 30, 31, 32] face three fundamental challenges, limiting their performance in real-world applications. First, training fully end-to-end OLLMs requires high-quality tri-modal datasets (images, speech, and text), which are scarce, expensive, and difficult to curate at scale. Most open-source models rely on true tri-modal corpora and ignore pairwise datasets (*e.g.*, image-text or speech-speech), resulting in suboptimal cross-modal alignment and weaker generalization. Without effective zero-shot alignment strategies, these models struggle to transfer learned representations across modalities, reducing their robustness in realistic multimodal tasks. Second, existing models predominantly rely on autoregressive (AR) architectures, which generate outputs sequentially, introducing high inference latency that hinders real-time multimodal interaction. Speech generation, in particular, is slow, as most models integrate external text-to-speech (TTS) modules [33], resulting in latency overhead and preventing end-to-end optimization. Achieving low-latency multimodal synthesis is essential for applications such as conversational AI, assistive technologies, and real-time interactive agents, where response time directly affects usability. Finally, emotionally expressive speech is critical for natural and engaging human-computer interactions. However, current OLLMs fail to generate emotionally consistent responses. Most models lack self-awareness, producing flat and robotic speech that does not modulate prosody, tone, or sentiment based on conversational context. Without direct preference optimization (DPO) [34, 35] for emotional speech, existing models struggle to align speech intonation with user emotions, leading to inauthentic and disconnected interactions. To summarize, these challenges significantly constrain the real-world applicability of open-source OLLMs, leaving commercial models far ahead in omnimodal reasoning, real-time interaction, and expressive speech synthesis.

To bridge this gap, we propose OpenOmni in this paper, which is a fully open-source two-stage training framework that enables efficient omnimodal learning while addressing the key limitations of existing models. As illustrated in Figure 1, OpenOmni introduces a progressive alignment strategy that enables cross-modal generalization from vision-language tasks to speech-language tasks, eliminating the need for expensive tri-modal datasets and computing resources. It further incorporates a lightweight and end-to-end speech decoder that facilitates parallel text and speech generation, which drastically reduces inference latency compared to autoregressive models. Moreover, by leveraging direct preference optimization (DPO), our model generates emotionally coherent and context-aware speech without requiring additional control modules or handcrafted prompts.

Extensive experiments confirm that OpenOmni achieves state-of-the-art performance in omnimodal alignment, real-time speech synthesis, and emotional speech generation. Compared to VITA [31], the leading fully open-source OLLM, which employs a 7×8B language model trained on 5M sam-

ples, OpenOmni attains superior results with a smaller model size (7B vs. 7×8B) and 3× fewer training samples (1.6M vs. 5M) while outperforming VITA by four absolute points on the OmniBench benchmark [36]. Additionally, OpenOmni reduces speech generation latency by 5×, achieving real-time inference (<1 second) and improving emotion classification accuracy by 7.7%. Our main contributions are summarized as follows:

- **High-quality speech datasets.** We construct O2S-300K and EO2S-9K, comprising 8,000 hours of bilingual text-synthesized speech, which enables efficient speech generation and emotional preference optimization.

- **Effective zero-shot omnimodal alignment.** We introduce a scalable and model-agnostic framework that enables low-resource and rapid omnimodal alignment using text as a pivot, followed by speech generation and emotional preference training. This approach allows the rapid development of an advanced all-modal assistant akin to GPT-4o.

- **An end-to-end omnimodal LLM.** We train OpenOmni with integrated text, image, and speech understanding progressively. After speech generation training and emotional preference optimization, OpenOmni can naturally produce real-time emotional speech.

## 2    Related Work

**Vision-language models.** The rapid advancement of vision-language models (VLMs) has been largely driven by the remarkable success of large language models (LLMs) [37, 38, 39, 40, 41] and the increasing availability of diverse image-text instruction data [18, 42, 43, 44, 45, 46, 47, 48]. LLaVA [18] and MiniGPT-4 [49] demonstrate strong cross-task generalization by integrating visual encoders with LLMs through lightweight connector modules trained on instruction datasets. To further enhance visual perception, LLaVA-NeXT [19] employs dynamic resolution techniques, which improve the adaptability to images of varying sizes and complexities. Expanding beyond conventional methods, DreamLLM [50] explores interleaved generation, enabling the simultaneous production of images and text within a shared multimodal context. Meanwhile, DEEM [21] enhances model robustness by employing diffusion models to extract visual features, which replaces traditional visual encoders and simplifies the overall architecture. These innovations collectively contribute to advancing vision-language reasoning in multimodal systems. Readers can refer to [51, 52, 53, 54, 55] for more details and recent advances in VLMs.

**Speech-language models.** Recent advancements in speech-language models (SLMs) [56, 57, 58, 59, 60] have significantly improved human-computer interactions by enabling direct speech processing without relying on intermediate text transcription. For example, SpeechGPT [11] and LLaMA-Omni [25] eliminate the need for explicit text-based transcriptions, reducing latency in multimodal content generation. For full-duplex dialogue systems, Moshi [61] and OmniFlatten [62] introduce mechanisms for handling simultaneous speech and text streams, adeptly managing challenges such as overlapping speech and interruptions [63]. Meanwhile, Freeze-Omni [64] introduces an innovative training approach that preserves the core capabilities of the original LLM, allowing low-latency speech-to-speech dialogue without requiring modifications to the pre-trained architecture. Focusing on emotional speech synthesis, Emo-DPO [65] applies direct preference optimization to generate expressive and controllable emotional speech, which addresses the emotional coherence gap in existing speech-language models. These developments mark a significant shift towards more natural and real-time speech interactions in multimodal AI systems. Readers can refer to [66] for more details of SLMs.

**Omnimodal language models.** With the development of multimodal research, models are increasingly shifting towards unified frameworks that seamlessly integrate diverse input and output modalities [67]. By tokenizing different data types into a shared representation, models like AnyGPT [29] and Unified-IO 2 [68] achieve seamless cross-modal task adaptability, allowing them to process audio, text, and images without significant architectural modifications. More recently, Mini-Omni2 [69] extends multimodal capabilities by integrating visual and auditory encoders, enabling real-time multimodal responses while incorporating mechanisms for detecting and interpreting semantic interruptions. Meanwhile, video-SALMONN [30] enhances video understanding by incorporating fine-grained temporal modeling, improving the model's ability to interpret speech

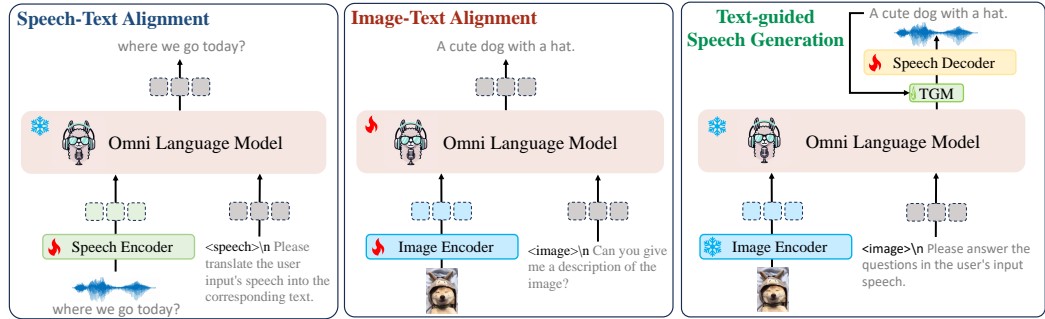

Figure 2: **Overview of the training process of OpenOmni.** To enable zero-shot omnimodal learning and real-time emotional speech generation, OpenOmni undergoes a progressive three-stage training process: (1) **Speech-text alignment.** A speech encoder extracts continuous speech and text features for alignment learning, equipping the large language model with speech understanding capabilities. (2) **Image-text alignment.** An image encoder extracts continuous image and text features, facilitating alignment learning that enhances OpenOmni's image comprehension and instruction-following abilities. This process also establishes implicit omnimodal alignment, which enables omni-understanding. (3) **Text-guided speech generation.** A lightweight speech decoder is trained using high-quality synthesized speech dialogue data, with a focus on direct preference optimization for emotional speech. This final stage allows OpenOmni to generate real-time and self-aware emotional speech. A text-guided module (TGM) is utilized to accelerate the training convergence.

and actions within videos. To enhance human-computer interaction, VITA [31] introduces duplex communication schemes, enabling fluid and intuitive exchanges between users and AI models. EMOVA [32] further extends the expressive capabilities of multimodal systems by integrating controllable emotional speech synthesis, which provides more natural and engaging user interactions.

Building upon these advancements, OpenOmni introduces a novel method for nearly zero-shot omnimodal alignment across language, vision, and speech, which incorporates self-aware emotional speech synthesis to enhance expressiveness and realism. By optimizing for speed, data efficiency, and generalization, OpenOmni achieves state-of-the-art performance in omnimodal tasks, surpassing previous models in real-time speech generation, multimodal alignment, and emotion-aware synthesis. Note that compared to Qwen-Omni [70], OpenOmni is a fully open-source solution focused on achieving advanced OLLMs under limited training and data resources, which helps researchers to easily conduct their studies and accelerate innovation in the field.

## 3 Method

In this section, we first formulate the omnimodal learning problem and provide an overview of the training procedure of OpenOmni as demonstrated in Figure 2. Afterward, we describe the specific training procedures for omnimodal alignment and real-time speech generation step by step.

### 3.1 Problem Setup and OpenOmni Overview

**Problem setup.** Omnimodal learning aims to model the relationships between images ($\mathbf{x}^{\mathrm{v}}$), speech ($\mathbf{x}^{\mathrm{s}}$), and text ($\mathbf{x}^{\mathrm{t}}$). The speech-to-text alignment task, which generates relevant text responses given input speech encoded by a speech encoder $h_{\mathrm{s}}(\cdot)$, is formulated as learning $p_{\boldsymbol{\phi}}(\mathbf{x}^{\mathrm{t}}|h_{\mathrm{s}}(\mathbf{x}^{\mathrm{s}}))$, parameterized by $\boldsymbol{\phi}$. Similarly, the image-to-text alignment task, which involves generating textual descriptions for input images encoded by an image encoder $h_{\mathrm{v}}(\cdot)$, is modeled as learning the conditional distribution $p_{\boldsymbol{\theta}}(\mathbf{x}^{\mathrm{t}}|h_{\mathrm{v}}(\mathbf{x}^{\mathrm{v}}))$, parameterized by $\boldsymbol{\theta}$. Lastly, the omnimodal-to-speech generation task, which synthesizes speech responses based on input text, speech, and images, is represented as learning $p_{\boldsymbol{\psi}}(\mathbf{x}^{\mathrm{s}}|f_{\mathrm{LLM}}(\mathbf{x}^{\mathrm{t}}, h_{\mathrm{s}}(\mathbf{x}^{\mathrm{s}}), h_{\mathrm{v}}(\mathbf{x}^{\mathrm{v}})))$, parameterized by $\boldsymbol{\psi}$, where $f_{\mathrm{LLM}}$ represents the large language model.

In the setting of standard omnimodal learning, training typically relies on image-speech-text triples $\mathcal{D}_\text{o} = \{(\mathbf{x}_i^\text{v}, \mathbf{x}_i^\text{s}, \mathbf{x}_i^\text{t})\}_{i=1}^K$. Nevertheless, high-quality image-text-speech datasets are scarce. To mitigate this limitation, we introduce text as a pivot, which leverages a large-scale speech-text dataset $\mathcal{D}_\text{s-t} = \{(\mathbf{x}_i^\text{s}, \mathbf{x}_i^\text{t})\}_{i=1}^M$ and image-text dataset $\mathcal{D}_\text{v-t} = \{(\mathbf{x}_i^\text{v}, \mathbf{x}_i^\text{t})\}_{i=1}^N$, where $M \gg K$ and $N \gg K$. Inspired by human learning mechanisms, where individuals naturally align visual concepts with speech across languages, OpenOmni transfers visual concepts learned from image-text tasks to speech understanding. Technically, OpenOmni decomposes the omnimodal alignment process into two consecutive stages: *speech-text alignment* and *image-text alignment*. The speech-text alignment stage establishes cross-modal alignment between speech $\mathbf{x}^\text{s}$ and text $\mathbf{x}^\text{t}$. This is achieved by training a speech LLM on text-speech pairs $\mathcal{D}_\text{s-t}$ with the objective $p_\phi(\mathbf{x}^\text{t}|h_\text{s}(\mathbf{x}^\text{s}))$, which also ensures that the hidden representations of semantically similar speech-text pairs are close. In the image-text alignment stage, OpenOmni utilizes image-text pairs $\mathcal{D}_\text{v-t}$ to optimize the objective $p_{\boldsymbol{\theta}}(\mathbf{x}^\text{t}|h_\text{v}(\mathbf{x}^\text{v}))$. Note that OpenOmni is architecture-agnostic, which allows flexible integration with existing advanced model architectures. Below, we detail OpenOmni.

## 3.2 Speech-Text Alignment

We incorporate a speech encoder $h_\text{s}$ to extract audio features from input speech $\mathbf{x}^\text{s}$. These audio features $h_\text{s}(\mathbf{x}^\text{s})$ are then replaced with corresponding text as input into the LLM. Following recent work to train speech conversation models [25, 24, 11], we pretrain OpenOmni on a large scale of text-speech pairs using the language modeling objective:

$$\mathcal{L}_\text{s-t}(\boldsymbol{\phi}, \mathcal{D}_\text{s-t}) = -\sum_{i=1}^M \log p_\phi(\mathbf{x}_i^\text{t}|h_\text{s}(\mathbf{x}_i^\text{s})). \tag{1}$$

## 3.3 Image-Text Alignment

We incorporate an image encoder $h_\text{v}$ to provide visual features as $h_\text{v}(\mathbf{x}^\text{v})$. These visual features are then concatenated with the text embedding as input into the speech LLM. Following prior work to train image-text conversation models [18, 42], OpenOmni's training process for image-text alignment consists of two sub-stages: image-text pretraining and image-text instruction tuning.

**Image-text pretraining.** In this sub-stage, we pretrain the visual module to align it with the LLM on a large scale of image-text pairs using the language modeling objective:

$$\mathcal{L}_\text{v-t}(\boldsymbol{\theta}, \mathcal{D}_\text{v-t}) = -\sum_{i=1}^N \log p_{\boldsymbol{\theta}}(\mathbf{x}_i^\text{t}|h_\text{v}(\mathbf{x}_i^\text{v})). \tag{2}$$

Here we fix the parameters of the LLM to prevent short texts in the image-text pairs from influencing the general capabilities.

**Image-text instruction tuning.** To enhance models' capabilities in following human instructions, we conduct instruction tuning on elaborately curated multimodal instruction tuning datasets built by blending existing image-text instruction tuning datasets. We denote this image-text instruction tuning dataset as $\mathcal{D}_\text{v-t}^\text{I} = \{(\mathbf{x}_i^\text{v}, \mathbf{x}_i^\text{t,q}, \mathbf{x}_i^\text{t,a})\}_{i=1}^L$, where $\mathbf{x}_i^\text{t,q}$ denotes the instruction and $\mathbf{x}_i^\text{t,a}$ is the response. Both the visual module and the speech LLM are then fine-tuned by maximizing the probability of the response:

$$\mathcal{L}_\text{v-t}^\text{I}(\boldsymbol{\theta}, \mathcal{D}_\text{v-t}^\text{I}) = -\sum_{i=1}^L \log p_{\boldsymbol{\theta}}(\mathbf{x}_i^\text{t,a}|h_\text{v}(\mathbf{x}_i^\text{v}), f_\text{LLM}(\mathbf{x}_i^\text{t,q})). \tag{3}$$

**Remark.** We observe a *quasi-zero-shot* transfer capability in OpenOmni within this scenario. When instruction tuning is performed exclusively on the image-text dataset, the model demonstrates the ability to respond accurately to an image $\mathbf{x}^{\mathrm{v}}$ and either a text-based question $\mathbf{x}^{\mathrm{t,q}}$ or an instruction provided in speech $\mathbf{x}^{\mathrm{s,q}}$. However, its responses are predominantly in text. This behavior can be attributed to the inherent similarity between the hidden representations of textual and spoken instructions learned by the LLM, *i.e.*, $f_{\mathrm{LLM}}(\mathbf{x}^{\mathrm{t,q}}) \approx f_{\mathrm{LLM}}(h_{\mathrm{s}}(\mathbf{x}^{\mathrm{s,q}}))$. Consequently, the model satisfies the following approximation: $p_{\boldsymbol{\theta}}(\mathbf{x}^{\mathrm{t,a}}|h_{\mathrm{v}}(\mathbf{x}^{\mathrm{v}}), f_{\mathrm{LLM}}(\mathbf{x}^{\mathrm{t,q}})) \approx p_{\boldsymbol{\theta}}(\mathbf{x}^{\mathrm{t,a}}|h_{\mathrm{v}}(\mathbf{x}^{\mathrm{v}}), f_{\mathrm{LLM}}(h_{\mathrm{s}}(\mathbf{x}^{\mathrm{s,q}})))$. OpenOmni completes the progressive omnimodal alignment, enabling the LLM to achieve a comprehensive understanding across image, text, and speech modalities.

## 3.4 Text-Guided Speech Generation

For speech generation, we incorporate a speech decoder $h_{\mathrm{s}}^{\mathrm{de}}$ to generate speech based on the output of the LLM $f_{\mathrm{LLM}}$. The speech generation training process in OpenOmni consists of two sub-stages: speech decoder training and emotional speech direct preference optimization (DPO).

**Speech decoder training.** To equip OpenOmni with real-time speech generation for enhancing interactive experiences, we adopt a streaming speech decoder, which supports both autoregressive (AR) and non-autoregressive (NAR) speech decoding modes. Besides, we curate a dataset, termed OpenOmni-300K, consisting of 300K single-round image-text instructions from MMEvol [42] and UltraChat [71] with corresponding speech responses for speech decoder training. We denote this dataset as $\mathcal{D}_{\mathrm{o\text{-}s}}^{\mathrm{I}} = \{(\mathbf{x}_i^{\mathrm{v}}, \mathbf{x}_i^{\mathrm{t,q}}, \mathbf{x}_i^{\mathrm{t,a}}, \mathbf{x}_i^{\mathrm{s,a}})\}_{i=1}^{L}$, where $\mathbf{x}^{\mathrm{s,a}}$ is the speech response.

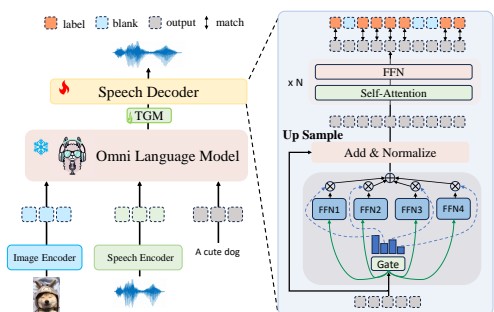

Figure 3: **The structure of our speech decoder.** The speech decoder consists of a mixture of expert modules and multiple transformer layers, which achieves end-to-end speech unit learning through the connectionist temporal classification (CTC) loss.

To process the speech response $\mathbf{x}^{\mathrm{s,a}}$, we follow [11, 25] to discretize speech into discrete units. Specifically, we use a pretrained speech tokenizer [11] to extract continuous speech representations and then convert these representations into a single unit, resulting in the final discrete unit sequence $\mathbf{x}^{\mathrm{u,a}} = [\mathrm{x}_1^{\mathrm{u,a}}, \mathrm{x}_2^{\mathrm{u,a}}, \dots, ]$, where $\mathrm{x}_i^{\mathrm{u,a}} \in \{0, 1, \dots, V-1\}$ with $V$ is the speech vocabulary size. The discrete units can then be converted back into a waveform using an additional unit-based vocoder [72], trained on English and Chinese datasets. As shown in Figure 3, we integrate the streaming speech decoder after the LLM to generate speech responses. The decoder consists of a *mixture of expert (MoE)* layer and a tiny standard decoder-only language model. The MoE layer stabilizes training and accelerates convergence—without this layer, the speech decoder fails to train effectively. Similar to [73, 74, 25], the speech decoder takes the output hidden states from the LLM as input and generates the discrete unit sequence corresponding to the speech response in real-time.

Given the output hidden states of the text response, denoted by $f_{\mathrm{LLM}}(\mathbf{x}^{\mathrm{v}}, \mathbf{x}^{\mathrm{s,q}})$, we first pass these hidden states through the text-guided module (TGM) to obtain the transformed hidden state $\mathbf{c}$. Then, $\mathbf{c}$ is fed into the speech decoder layers, leading to the final hidden state sequence $\mathbf{o}$. We use *connectionist temporal classification (CTC)* [75] to align $\mathbf{o}$ with the discrete unit sequence $\mathbf{x}^{\mathrm{u,a}}$. During training, CTC marginalizes over all possible alignments as follows:

$$\mathcal{L}_{\mathrm{CTC}}(\boldsymbol{\psi}, \mathcal{D}_{\mathrm{o\text{-}s}}^{\mathrm{I}}) = -\log p_{\boldsymbol{\psi}}(\mathbf{x}^{\mathrm{u,a}}|\mathbf{o}) = -\log \sum_{\mathbf{A} \in \Delta(\mathbf{x}^{\mathrm{u,a}})} p_{\boldsymbol{\psi}}(\mathbf{A}|\mathbf{o}) = -\log \sum_{\mathbf{A} \in \Delta(\mathbf{x}^{\mathrm{u,a}})} \prod p_{\boldsymbol{\psi}}(\mathrm{x}_i^{\mathrm{u,a}}|\mathbf{o}),$$

(4)

where $\Delta(\mathbf{x}^{\mathrm{u,a}})$ denotes all possible alignments that collapse to $\mathbf{x}^{\mathrm{u,a}}$. During inference, the best alignment is selected as $\mathbf{A}* = \arg\max_{\mathbf{A}} p(\mathbf{A}|\mathbf{o})$. The corresponding discrete unit sequence is fed into the vocoder to synthesize the waveform.

**Emotional speech DPO.** To enable OpenOmni to generate self-aware, emotionally coherent, and expressive speech based on contextual history without additional control modules, we introduce

the *CTC-DPO* algorithm. This method enhances smooth and natural dialogue interactions and is formulated as

$$\mathcal{L}_{\text{CTC-DPO}} = -\mathbb{E}_{(\mathbf{x}, \mathbf{y}_{\text{w}}, \mathbf{y}_{\text{l}})}[\log \sigma(\beta \log \frac{\pi_*(\mathbf{y}_{\text{w}}|\mathbf{x})}{\pi_{\text{ref}}(\mathbf{y}_{\text{w}}|\mathbf{x})} - \beta \log \frac{\pi_*(\mathbf{y}_{\text{l}}|\mathbf{x})}{\pi_{\text{ref}}(\mathbf{y}_{\text{l}}|\mathbf{x})})], \tag{5}$$

where $\beta$ is a constant, $\sigma$ is the sigmoid function, and $(\mathbf{y}_{\text{w}}, \mathbf{y}_{\text{l}})$ is the preference pair. Besides, the reference model $\pi_{\text{ref}}$ is the pretrained model from the real-time speech generation stage and remains fixed during DPO training. Only the policy model $\pi_*$ is updated. Compared to traditional reinforcement learning with human feedback (RLHF) [76], the DPO paradigm is simpler, more efficient, and more stable for aligning OpenOmni with self-aware emotional speech generation.

Following the Plutchik model of emotions [77], we construct a multi-turn dialogue preference dataset incorporating nine distinct emotions. Each preference pair consists of an emotionally congruent speech response unit sequence $\mathbf{y}_{\text{w}} = \mathbf{x}_{\text{w}}^{\text{u,a}}$, which aligns with the conversational history, and an emotionally neutral sequence $\mathbf{y}_{\text{l}} = \mathbf{x}_{\text{l}}^{\text{u,a}}$, which is inconsistent with the context. The policy model $\pi_*$ during training is optimized as: $-\log \pi_*(\mathbf{y}|\mathbf{x}) = -\log \sum_{\mathbf{A} \in \Delta(\mathbf{x}^{\text{u,a}})} \prod p_\psi(\mathbf{x}_i^{\text{u,a}}|\mathbf{o})$. After training, OpenOmni is capable of generating real-time and emotionally expressive multi-turn dialogues.

# 4 Experiments

## 4.1 Implementation Details

In this subsection, we introduce data construction and the models used. More details about the data and optimization strategy of OpenOmni can be found in Appendix A and Appendix D respectively.

**Omnimodal alignment data.** During the speech-text alignment phase, in addition to WeNet-Speech [78], LibriSpeech [79], and AIShell-4 [80], we exploit portions of shorter responses from O2S-300K, totaling 8,000 hours of data, for bilingual speech-text alignment training. For image-text alignment, we train OpenOmni on the LLaVA-Pretrain-595K [18]. Besides, in the image-text instruction tuning stage, we fine-tune OpenOmni on the compact high-quality dataset MMEvol [42] for efficient optimization.

**Speech decoder training data.** To support real-time speech generation, we curate a dataset of 300K instructions from MMEvol [42] and UltraChat [71] that include long responses for training the speech decoder. Specifically, we decompose multi-turn dialogues into single-turn question-answer pairs, rank the responses based on their length, and select 100K question-answer pairs with relatively long responses. To support bilingual output in Chinese and English, we translate 50K question-answer pairs into their corresponding Chinese versions using GPT-4o-mini API, and then convert the answers into the corresponding speech using CosyVoice [33]. We employ the same method for text-conditioned speech synthesis on 200K randomly selected data from UltraChat. As a result, we obtain 8,000 hours of high-quality bilingual speech generation data, termed O2S-300K.

**Emotional speech DPO data.** Based on the Plutchik model of emotions [77], which categorizes emotions into eight distinct types, we curate a multi-turn speech preference dataset, EO2S-9K, for self-awareness emotion evaluation. In more detail, we randomly select 200K samples from MMEvol and employ Qwen2-72B [2] to categorize responses into nine predefined emotions per round. From this, we extract 1K bilingual dialogues labeled with emotion categories, reserving an additional 100 samples as an emotional test set for evaluating self-aware speech generation. Since certain emotions, such as anger and sadness, are underrepresented in the MMEvol dataset, we augment the dataset using the GPT-4o-mini API to ensure sufficient data for these categories. The final dataset maintains an equal representation of Chinese and English samples. To further enhance emotional preference training, we use CosyVoice [33] to generate unconditional speech as negative samples and emotion-conditioned speech as positive samples, constructing preference pairs for training direct preference optimization in emotional speech generation.

**Models.** We design the architecture following LLaVA series [18, 19], where the omnimodal large language model consists of four key components: an LLM (Qwen2.5-7B-Instruct [2]) for next token prediction, an image encoder (CLIP-ViT-L [81]) for extracting visual features, a speech encoder

Table 1: **Overall omni-understanding results on OmniBench.** In each case, the best result is indicated in **bold**, and the second-best result is marked with an underline.

| Method | Action & Activity | Story Description | Plot Inference | Identification & Description | Contextual & Environmental | Identity & Relationship | Text & Symbols | Count & Quantity | Overall |
|---|---|---|---|---|---|---|---|---|---|
| AnyGPT (7B) [29] | 5.98 | 8.70 | 7.59 | 4.74 | 5.67 | 12.50 | 8.00 | 20.00 | 7.01 |
| Video-SALMONN (13B) [30] | 28.69 | 25.65 | 24.47 | 23.22 | 29.08 | 21.83 | **52.00** | 26.63 | 26.53 |
| UnifiedIO2-Large (1.1B) [68] | 28.29 | 22.17 | 32.49 | 30.81 | 28.37 | 21.83 | 16.00 | 13.33 | 27.76 |
| UnifiedIO2-XLarge (3.2B) [68] | 30.28 | 26.52 | 30.38 | 31.75 | 28.37 | 18.75 | 28.00 | 26.63 | 29.16 |
| UnifiedIO2-XXLarge (6.8B) [68] | 27.49 | 23.04 | 28.69 | 25.59 | 26.95 | 12.50 | 12.00 | **46.67** | 25.92 |
| Baichuan-Omni (7B) [84] | - | - | - | - | - | - | - | - | 33.25 |
| VITA (7×8B) [31] | 33.47 | 34.35 | 27.00 | 36.02 | 43.97 | 31.25 | 24.00 | 6.67 | 33.45 |
| VITA-1.5 (7B) [31] | - | - | - | - | - | - | - | - | 33.48 |
| OpenOmni (7B) | **36.65** | **45.65** | **32.91** | **44.08** | **48.23** | **34.38** | 24.00 | 33.33 | **37.40** |

Table 2: **Comparison with state-of-the-art methods on visual-language benchmarks**. This includes an indication of audio input/output support. The best performance among fully open-source models is highlighted in **bold**.

| Model | w/ Audio IO | PT | IT | MMStar | MMB | MMB$^{CN}$ | HallBench | MathVista$^M$ | MMMU$^V$ | AI2D | RWQA |
|---|---|---|---|---|---|---|---|---|---|---|---|
| **Proprietary Models** | | | | | | | | | | | |
| GPT-4o | ✓ | – | – | - | 83.4 | 82.1 | 55.0 | 63.8 | 69.1 | - | 75.4 |
| GPT-4o-mini | ✓ | – | – | - | - | - | 46.1 | 52.4 | 60.0 | - | 67.1 |
| **Weight Open-Source** | | | | | | | | | | | |
| MiniCPM-V2.5 (8B) [85] | ✗ | 570M | 9.1M | 51.3 | 76.7 | 73.3 | 42.5 | 54.3 | 45.8 | - | 63.5 |
| Qwen2-VL-Chat (7B) [20] | ✗ | 1.4B | - | 60.7 | 86.4 | 81.9 | 50.6 | 58.2 | 52.7 | - | 69.7 |
| Baichuan-Omni (7B) [84] | ✓ | – | 8M | - | 76.2 | 74.9 | 47.8 | 51.9 | 47.3 | - | 62.6 |
| EMOVA (8B) [32] | ✓ | 7.4M | 4.4M | - | 82.8 | - | - | 61.1 | - | 82.8 | 64.3 |
| **Fully Open-Source** | | | | | | | | | | | |
| Cambrain-I (8B) [86] | ✗ | 2.5M | 7M | 50.7 | - | - | 34.3 | 47.0 | 41.8 | 73.1 | 64.2 |
| MMEvol (7B) [42] | ✗ | 0.6M | 1.5M | 51.6 | 74.6 | 74.3 | 42.9 | 52.4 | 45.1 | 74.7 | 63.9 |
| VITA (7×8B) [31] | ✓ | – | 5M | - | 74.7 | 71.4 | 39.7 | 44.9 | 45.3 | 74.3 | 59.0 |
| OpenOmni (7B) | ✓ | 0.6M | 1.7M | **52.3** | 76.2 | 76.4 | **44.2** | 52.7 | 46.7 | 74.8 | 64.3 |

(Whisper-large-v3 [82]) for extracting audio features and a streaming speech decoder (Qwen2.5-0.5B-Instruct [2]) for generating vivid speech in real-time. Moreover, an image-text projector and a speech-text projector are adopted to align the image-text and speech-text modalities, respectively. The MoE module and the text-guided module are designed to align the omnimodal embedding and speech decoder efficiently and stably. For the autoregressive mode, we use the speech tokenizer from GLM4-Voice [83] with a vocabulary size of 16K, which leads to better speech quality. For non-autoregressive models, we use the CosVoice [33] speech tokenizer with a smaller vocabulary size of 6K, which facilitates faster convergence during CTC-based optimization. All experiments are conducted on 8×NVIDIA A100-80G GPUs.

## 4.2 Main Results and Discussions

**Omni-language evaluation.** OmniBench [36] is a pioneering benchmark designed to evaluate omnimodal large language models (OLLMs) by assessing their ability to integrate and interpret simultaneous inputs from images, audio, and text. This evaluation framework consists of 1,142 question-answer pairs categorized into tasks that focus on cognitive and reasoning abilities, which poses significant challenges in entity recognition, causal inference, and abstract concept comprehension. We compare our OpenOmni with other OLLMs on OmniBench, with results summarized in Table 1. Notably, our model achieves excellent zero-shot omnimodal alignment using only two training phases: speech-text alignment and image-text alignment. Compared to the fully open-source state-of-the-art OLLM, *e.g.*, VITA [31], which is trained on tri-modal data (image-speech-text triplets), OpenOmni achieves superior overall results on OmniBench (37.40 vs. 33.45) despite using significantly fewer training parameters (7B vs. 7×8B) and less image-text training data (1.6M vs. 5M). Furthermore, by leveraging text as a pivot, our method completes omnimodal alignment implicitly, which demonstrates enhanced scalability in scenarios with limited tri-modal data. In addition to OmniBench, we provide empirical results on AV-Odyssey Bench in Appendix B.1.

**Vision-language evaluation.** To assess the effectiveness of OpenOmni in aligning image-text modalities, we compare its performance against previous vision-language models (VLLMs) across eight representative benchmarks: MMBench-EN [87], MMBench-CN [87], MMStar [88], RealWorldQA [89], MMMU [90], MathVista [91], AI2D [92], and HallusionBench [93]. To en-

Table 3: **Comparison with state-of-the-art methods on speech-language benchmarks.** In each case, the best result among Omnimodal LLMs is indicated in **bold**.

| Model | AIShell-2 (ZH-CER) | | | | Librispeech (EN-WER) | | | |
| | Dev | | Test | | Test_clean | | Test_other | |
| | S2T | T2S | S2T | T2S | S2T | T2S | S2T | T2S |
|---|---|---|---|---|---|---|---|---|
| **Speech LLM** | | | | | | | | |
| SpeechT5 [95] | - | - | - | - | 2.4 | - | 5.8 | - |
| SALMONN [96] | - | - | - | - | 2.1 | - | 4.9 | - |
| Mini-Omni (7B) [69] | - | - | - | - | 4.7 | - | 9.4 | - |
| Freeze-Omni (7B) [64] | - | - | - | - | 3.2 | - | 7.7 | - |
| Qwen2-Audio (7B) [24] | 3.1 | - | 3.3 | - | 2.0 | - | 4.5 | - |
| **Omnimodal LLM** | | | | | | | | |
| AnyGPT (13B) [29] | - | - | - | - | 8.5 | - | - | - |
| VITA (7×8B) [31] | - | - | - | - | 8.1 | - | 18.4 | - |
| EMOVA (7B) [32] | 10.3 | 7.9 | - | - | 4.0 | 3.4 | - | - |
| VITA 1.5 (7B) [31] | - | - | - | - | 3.4 | - | 7.5 | - |
| OpenOmni (7B) | **6.8** | **7.3** | **6.9** | **13.1** | **3.1** | **2.6** | **4.1** | **5.6** |

Table 4: **Overall self-aware emotional speech generation results on the bilingual EO2S-9K test set.** In each case, the best result is indicated in **bold**.

| Model | Lang | Angry & Disgusted | Fearful | Happy | Neutral | Other | Sad | Surprised | Overall |
|---|---|---|---|---|---|---|---|---|---|
| OpenOmni | ZH | 89.7 | 54.8 | 33.3 | 92.3 | 51.6 | 60.2 | 23.7 | 57.9 |
| w/ DPO | ZH | **96.6** | **78.4** | **37.7** | **97.1** | **62.8** | **90.7** | **29.8** | **70.4** |
| OpenOmni | EN | 89.2 | 68.7 | 57.5 | 91.9 | 48.0 | 75.6 | 7.5 | 62.6 |
| w/ DPO | EN | **91.3** | **70.4** | **60.6** | **94.6** | **49.6** | **77.3** | **13.9** | **65.4** |

sure reproducibility and maintain consistency across all models and benchmarks, we employ VLMEvalKit [94] for zero-shot evaluation. As shown in Table 2, OpenOmni achieves superior results compared to the fully open-source state-of-the-art OLLM, VITA [31], despite being trained on significantly less data. Notably, our model outperforms VITA with gains of 7.0% on MMBench-Chinese and 11.3% on HallusionBench. We can also observe that the use of additional speech modals can further enhance the vision-language capabilities of the model. Furthermore, compared to other fully open-source VLMs, OpenOmni maintains competitive performance despite reduced training data, which demonstrates the effectiveness of our image-text alignment strategy.

**Speech-language evaluation.** To evaluate the speech understanding and generation capabilities of our OpenOmni, we measure word error rate (WER) on the AIshell-2 [80] and Librispeech [79] benchmarks for two tasks: speech-to-text recognition (S2T) and text-to-speech generation (T2S). Specifically, for T2S evaluation, we use Whisper-large-V3 to transcribe OpenOmni's synthesized speech and compute WER against ground-truth text labels. As shown in Table 3, OpenOmni achieves the best performance on both S2T and T2S tasks for bilingual (Chinese and English) data, and outperforms other omnimodal models. These results indicate that OpenOmni not only comprehends speech effectively but also generates fluent and high-quality audio while maintaining strong alignment between speech and text modalities. Additionally, compared to VITA [31], which relies on separate text-to-speech (TTS) models, and EMOVA [32], which uses an autoregressive (AR) structure, OpenOmni demonstrates significantly faster speech generation via two-mode support. Owing to its end-to-end, lightweight, and non-autoregressive (NAR) decoding mode, OpenOmni can generate up to 30 seconds of speech with less than one second of latency, which achieves real-time speech generation at over $5\times$ speed of autoregressive models.

**Emotional speech synthesis evaluation.** To assess the effectiveness of direct preference optimization in emotional speech generation, we evaluate OpenOmni's self-aware emotional speech synthesis on the EO2S-9K test set. Specifically, we use Emotion2Vec [97] to classify the emotions in the generated speech and measure accuracy against ground-truth labels. As shown in Table 4, direct preference optimization for emotional speech effectively enhances OpenOmni's ability to generate emotionally expressive speech. This improvement is particularly evident in bilingual and multi-

turn emotional speech generation tasks, demonstrating the model's ability to produce natural and contextually aware speech with accurate emotional intonation.

We also provide ablation studies to investigate the text-guided module (TGM), the number of layers and experts in the speech decoder, training strategy in alignment stages, and the performance of order in progressive alignment. Due to the limited page, experimental results and the following discussions can be checked in Appendix C.

## 5 Conclusion

In this paper, we introduce OpenOmni, a novel omnimodal model that leverages text as a pivot to achieve tri-modal zero-shot alignment, which addresses the challenge of limited tri-modal data. By integrating a lightweight streaming speech decoder with direct preference optimization for emotional speech, OpenOmni enables real-time, self-aware, and high-quality speech interactions. The extensive evaluations demonstrate that OpenOmni achieves state-of-the-art performance on multiple benchmarks while using significantly fewer training parameters and less training data than previous advanced models. Comprehensive ablation studies and discussions are also presented to rigorously validate our claims.

## Acknowledgements

This work was supported by Alibaba Research Intern Program. Min Yang is supported by Guangdong Basic and Applied Basic Research Foundation (2025B1515020032 and 2024A1515030166). Xiaobo Xia is supported by MoE Key Laboratory of Brain-inspired Intelligent Perception and Cognition, University of Science and Technology of China (Grant No. 2421002).

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

# A  Data Construction

We provide details of the data construction for multiple training stages below:

- **OpenOmni-1-1:** In addition to datasets WeNetSpeech, LibriSpeech, and AIShell-4, we randomly select 80K image-text instruction data with shorter responses from MMEvol [42]. We translate 40K of this data into Chinese using Qwen72B and synthesize the responses into speech data with CosVoice. This results in 1,600 hours of OpenOmni-1-1 data for speech-text alignment pretraining.
- **OpenOmni-2-1:** For rapid image-text alignment pretraining, we use the llava pretrain dataset, following previous work [18, 19, 42, 98].
- **OpenOmni-2-2:** To achieve efficient image-text instruction tuning, we employ MMEvol data. Since we later train the speech decoder by freezing the LLM mode, we include O2S-300K to stabilize the training of the speech decoder, leading to a combined dataset of 1.7M for OpenOmni-2-2.
- **OpenOmni-3-1:** To better utilize computational resources, we select 300K data with long response instructions from MMEvol and UltraChat. This includes 100K image-text instruction data, 100K single-round dialogue, and 100K multi-round dialogue. We synthesize the corresponding speech using CosVoice, resulting in 8,000 hours of O2S-300K.
- **OpenOmni-3-2:** We curate 9K emotion preference data and generate emotional speech preference pairs using CosVoice's conditional control. This is used for emotional speech direct preference optimization.

# B  Additional Experiments

## B.1  Additional Omni-Language Evaluation

In addition to OmniBench, we conduct experiments on the AV-Odyssey Bench [99], which involves the four modalities: audio, text, image, and video. For video, we test by averaging 8 sampled frames into a single image. The experimental results are shown in Table 5 below. Compared to other OLLMs, OpenOmni achieves the best average performance using only bi-modal speech-text and image-text data. With 7B model parameters and no audio or video training, it outperforms VITA by 4.4 points, demonstrating the effectiveness and efficiency of OpenOmni.

Table 5: **Overall omni-understanding results on AV-Odyssey Bench.** In each case, the best result is indicated in **bold**. We conduct a performance comparison of omni-understanding among various fully open-source Omnimodal Large Language Models (OLLMs) on AV-Odyssey Bench. Compared to the state-of-the-art OLLM, VITA [31], which was trained on tri-modal data, OpenOmni achieves comparable advanced performance using significantly less training data and smaller model size.

| Method | Timbre | Tone | Melody | Space | Time | Hall | Intricacy | Overall |
|---|---|---|---|---|---|---|---|---|
| OneLLM (7B) [100] | 25.0 | 25.5 | 21.5 | 37.5 | **29.3** | 25.5 | 38.4 | 27.4 |
| PandaGPT (7B) [101] | 23.5 | 23.2 | 27.6 | 45.0 | 23.8 | 28.0 | 23.9 | 26.7 |
| Video-LLaMA (7B) [102] | 25.5 | 22.3 | 24.4 | 30.0 | 26.2 | 25.0 | 30.7 | 26.1 |
| Video-LLaMA2(7B) [103] | 24.1 | 25.5 | 26.4 | 30.0 | 27.2 | **33.0** | 34.5 | 26.8 |
| AnyGPT (7B) [29] | 24.6 | 25.0 | 26.4 | 27.5 | 29.2 | 29.0 | 25.7 | 26.1 |
| NexTGPT (7B) [104] | 23.3 | 20.9 | 27.8 | 30.0 | 28.8 | 28.5 | 23.6 | 25.5 |
| VITA (7×8B) [31] | 24.1 | 26.4 | 27.8 | 22.5 | 26.3 | 31.0 | 36.8 | 26.4 |
| OpenOmni (7B) | 23.9 | **27.7** | 25.9 | **60.0** | 25.2 | 29.5 | **37.6** | **32.8** |

# C  Additional Ablation Studies

**On TGM.** To explore the effect of TGM on speech generation in two modes, we plot the change of training loss under the same setting. As shown in Figure 4, we can observe that TGM can significantly improve the convergence speed of training and the performance of model speech generation,

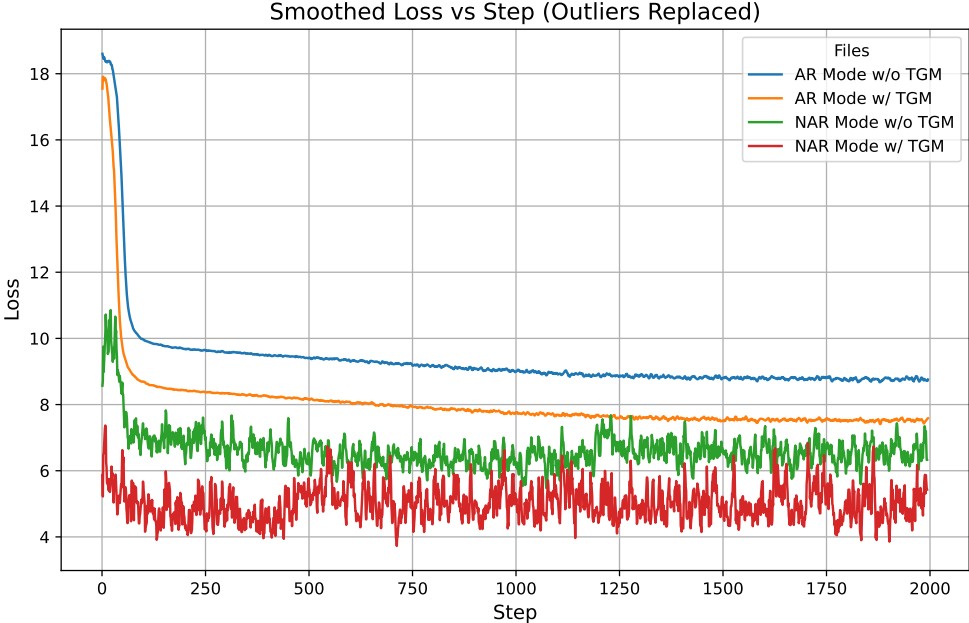

Figure 4: **Ablation study of the text-guided module (TGM).** In order to explore the effect of TGM on speech generation under the two modes, we plot the change of training loss under the same setting. TGM can significantly improve the convergence speed of training and improve the effect of speech generation of the speech decoder.

Table 6: **Ablation study on the number of layers and experts in the speech decoder.** Increasing experts in the mixture of experts module stabilizes the CTC loss during training and enhances speech generation capacity. Deeper transformer layers improve English and Chinese speech generation, with greater benefits for Chinese.

| Layers | Experts | Wenetspeech(ZH) | | Librispeech(EN) | |
|---|---|---|---|---|---|
| | | Test_Net | Test_Meeting | Test_clean | Test_other |
| 2 | 1 | 113.6 | 129.7 | 87.8 | 96.5 |
| 2 | 2 | 16.7 | 22.3 | 10.7 | 14.6 |
| 2 | 4 | 8.5 | 8.4 | 4.2 | 4.7 |
| 4 | 4 | 7.3 | 7.9 | **3.8** | **4.3** |
| 6 | 4 | **6.4** | **6.7** | 4.1 | 4.5 |

Table 7: **Ablation study of the model training in image-text alignment and speech-text alignment stages.** The speech and text have clear temporal correspondence, enabling low-cost alignment. In contrast, the image-text gap is larger, requiring LLM fine-tuning for better results.

| Stage | LLM freeze | GPUxHour | MMStar | MathVista[M] | MMMU[V] | AI2D |
|---|---|---|---|---|---|---|
| image-text | ✓ | 76 | 41.2 | 42.3 | 35.5 | 54.3 |
| image-text | ✗ | 192 | **44.4** | **47.6** | **40.2** | **59.1** |

| Stage | LLM freeze | GPUxHour | AIShell-2-Dev | AIShell-2-Test | LibriSpeech-Test-Clean | LibriSpeech-Test-Other |
|---|---|---|---|---|---|---|
| speech-text | ✓ | 32 | 12.7 | 11.5 | 9.8 | 13.3 |
| image-text | ✗ | 192 | 13.1 | 11.8 | 10.1 | 13.5 |
| speech-text | ✗ | 84 | **12.2** | **11.1** | **9.2** | **12.8** |

which verifies the effectiveness of our model design, whether it is a next-token-prediction (NTP) loss under the stable AR mode or CTC loss under the unstable NAR mode.

**On the number of layers and experts in the speech decoder.** To explore the impact of the number of layers in the NAR speech decoder and the MoE module on Chinese and English speech genera-

Table 8: **Ablation study of the alignment order and joint training strategy.** The order of the alignment strategies has minimal impact on the final performance. Compared to joint training, the multi-stage alignment strategy not only significantly reduces memory requirements during training but also ensures competitive results, making it the most efficient and optimal training strategy under low-resource conditions.

| Order First | Joint Training | VRAM | MMStar | HallBench | MathVista$^M$ | MMMU$^V$ | AI2D | RWQA | AIShell-2 | | LibriSpeech | |
| --- | --- | --- | --- | --- | --- | --- | --- | --- | --- | --- | --- | --- |
| | | | | | | | | | Dev | Test | Test-Clean | Test-Other |
| image-text | × | 40GB×8 | 44.7 | 35.9 | 47.1 | **40.7** | 58.6 | 60.1 | 13.4 | 11.3 | 10.4 | 13.6 |
| speech-text | × | 40GB×8 | 44.4 | 36.7 | 47.6 | 40.2 | 59.1 | 55.9 | 12.7 | 11.5 | 9.8 | 13.3 |
| speech-text | ✓ | 90GB×8 | **44.9** | **37.1** | **47.8** | 40.6 | **59.6** | **60.4** | **12.4** | **11.1** | **9.4** | **13.1** |

tion, we conduct ablation experiments on WeNetSpeech [78] and LibriSpeech [79]. As illustrated in Table 6, the instability and fragility associated with training using the CTC loss function present significant challenges. When simply employing a single feed-forward network (*i.e.*, the number of experts is 1), it becomes increasingly difficult to reconcile the conflicting training dynamics inherent in mixed-language scenarios, particularly when dealing with varying response lengths. As a result, training the speech decoder under these conditions proves to be quite challenging. Our findings demonstrate that incrementally increasing the number of experts significantly enhances the model's performance in bilingual speech generation, thereby underscoring the effectiveness of our MoE module design. However, we observe inconsistent preferences regarding the optimal number of layers in the speech decoder for generating speech in Chinese and English. Specifically, while four layers yield the best results for English generation, six layers are more suitable for generating Chinese speech.

**LLM training in image-text alignment and speech-text alignment stages.** To investigate whether training large language models affects modality alignment at different stages, we conduct an ablation study. As shown in Table 7, since speech and text data naturally have a temporal alignment relationship, freezing the LLM during alignment training still achieves competitive alignment performance. However, the gap between image and text modalities is significantly larger, and better alignment results are only achieved by unfreezing the LLM during training. Furthermore, we find that even after image-text training, there is no catastrophic forgetting of knowledge related to speech-text alignment. This validates the effectiveness and efficiency of our progressive alignment method.

**Alignment order and joint training strategy.** We conduct ablation studies to explore the impact of multi-stage alignment order and joint training strategies. By using 20K speech data and 500K image data, as shown in Table 8, we observe that the relative order of speech-text alignment and image-text alignment has little effect on the final performance, which indicates a low correlation between the two stages.

Due to limitations in data and computational resources, we adopt a multi-stage progressive multimodal alignment strategy to complete the omnimodal alignment training. At any given stage, only two modalities of data are processed simultaneously. This method not only alleviates the challenges posed by missing tri-modal data but also significantly reduces computational memory requirements. With fewer computational resources and less training data, our method achieves superior omnimodal alignment results compared to existing approaches.

As shown in Table 8, it can be observed that multi-stage training requires only 40GB×8 of VRAM, which is significantly lower than the memory demands of joint training. At the same time, it achieves comparable results, making it a more efficient and practical choice in resource-constrained scenarios.

# D Additional Implementation Details

OpenOmni is trained in five sequential sub-stages. Further details on these training stages are provided in Table 9.

Besides, as shown in Figure 5, we provide more details of the speech decoder design and training here. For the speech decoder, OpenOmni supports both autoregressive (AR) and non-autoregressive (NAR) methods. Specifically, the AR mode has better generation quality but a slower generation

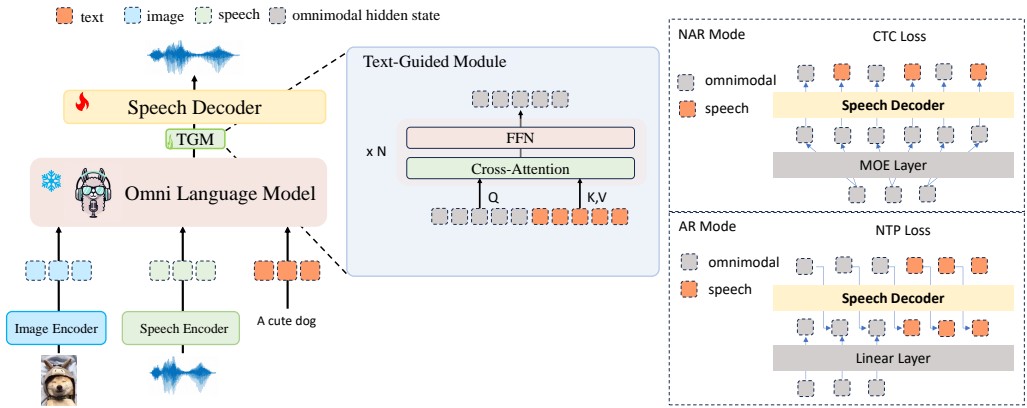

Figure 5: **Overview of text-guided module and speech decoder mode.** (*Left*) Text-guided module fuses the hidden state and response textual feature via cross-attention, accelerating convergence speed of training without dropping the speed of speech decoding and context emotion perception. (*Right*) OpenOmni supports both autoregressive (AR) and non-autoregressive speech (NAR) generation. The NAR mode uses the CTC loss modeling and a 6K speech vocabulary size to enable real-time parallel speech decoding generation. The AR mode uses the NTP loss modeling and a speech vocabulary size of 16K to support streaming decoding and higher-quality speech generation. To make the training of the speech generator more stable, we design a text-guided output feature fusion method to ensure the correctness of semantic alignment in speech generation modeling.

Table 9: The detailed training setup for OpenOmni and the hyperparameters across the training stage. All experiments are conducted on 8×NVIDIA A100-80G GPUs. The dataset index can be checked in Appendix A.

| Hyperparameter | I | II | III | IV | V |
|---|---|---|---|---|---|
| batch size | 256 | 128 | 128 | 32 | 32 |
| lr | $1 \times 10^{-3}$ | $1 \times 10^{-3}$ | $5 \times 10^{-5}$ | $5 \times 10^{-4}$ | $5 \times 10^{-4}$ |
| warmup ratio | 0.3 | 0.3 | 0.3 | 0.3 | 0.3 |
| epoch | 1 | 1 | 1 | 3 | 3 |
| freeze LLM | ✔ | ✔ | ✘ | ✔ | ✔ |
| optimizer | AdamW | AdamW | AdamW | AdamW | AdamW |
| cost | 40 GPU Hours | 80 GPU Hours | 500 GPU Hours | 36 GPU Hours | 8 GPU Hours |
| dataset | 1-1 | 2-1 | 2-2 | 3-1 | 3-2 |
| loss | $\mathcal{L}_{\text{s-t}}$ | $\mathcal{L}_{\text{v-t}}$ | $\mathcal{L}_{\text{v-t}}^{\text{I}}$ | $\mathcal{L}_{\text{CTC}}$ | $\mathcal{L}_{\text{CTC-DPO}}$ |

speed, while the NAR mode can achieve real-time speech generation, but the generation quality is slightly worse. At the same time, in order to train the speech generator more efficiently, we also design a text-guided feature fusion module, so that the conditional features used for speech generation have more accurate alignment semantics, which can improve the generation quality and training efficiency of the speech decoder.

**NAR mode.** In the NAR mode, the conditional features generated by OLLM are fed into the speech decoder by a layer of MoE and then upsampled to obtain the predicted speech output, and finally, the end-to-end optimization is carried out by the CTC loss modeling of the speech output. Due to the instability of CTC loss training, the smaller the size of the speech vocabulary, the easier it is to be successfully trained, but the generation quality of the corresponding speech will be affected by the smaller vocabulary.

**AR mode.** The AR mode projects the conditional features generated by OLLM into the speech space through a layer of linear layer and feeds them into the speech decoder to obtain the speech prediction output, and finally optimizes the speech output end-to-end by modeling the NTP loss. Due to the stability of NTP loss training, the quality of speech generation will be higher than that of NAR generation, but the speed of speech generation will be reduced by AR decoding.

Note that both AR and NAR modes depend on the quality of the speech generation conditional features generated by OLLM. Although OpenOmni will let the OLLM fit the text answer corresponding to the speech through multiple rounds of training in advance, there will still be OLLM outputs decoded into the wrong text answer. In this case, the erroneously generated condition features will be incorrectly aligned with the speech during the training process, which will ultimately reduce the performance of the speech decoder. To ensure the efficiency of training, OpenOmni fuses the speech generation condition features output by OLLM with the corresponding text features with correct semantics, and then feeds them into the speech decoder for speech generation modeling training. Through the feature fusion module of text prior, OpenOmni avoids the misalignment of speech and corresponding text and ultimately makes the speech decoder training more stable. At the same time, it enjoys more efficient and accurate speech generation quality.

# E    Broader Impacts

OpenOmni marks a significant advancement in open-source omnimodal large language models (OLLMs), seamlessly integrating vision, language, and speech into a unified framework. Its open-source nature fosters transparency, community-driven innovation, and trust in AI technologies. However, challenges remain, including emotional manipulation and privacy concerns. To address these, we emphasize responsible AI practices and secure handling of speech data.

# F    Limitation

Due to resource limitations, our method primarily focuses on Chinese and English and has not been trained or validated on multilingual data. In the future, we plan to utilize multilingual speech data to activate the multilingual capabilities of the large language model, thereby enhancing its applicability across diverse scenarios.

# G    Prompt Template

We provide the specific prompts used in our tasks, as well as examples of synthesized multi-turn emotional speech data, to help readers better understand this work.

---

**English Prompts used in Text-to-Speech (T2S) and Speech-to-Text (S2T) tasks.**

**S2T (ASR)**
Please translate the user's spoken input into corresponding text, word by word.
**T2S**
Echo game: Repeat this sentence back to me exactly as I said it!
Mirror challenge: Reflect my words right back at me!
Parrot mode: Can you mimic what I just said?
Copycat test: Let's see how well you can replicate my sentence!
Playback time: Rewind and repeat my last sentence.
Echo chamber: Say back to me what I just mentioned.
Simulate my speech: Can you say exactly what I just did?
Mimicry game: Echo my words precisely!
Repeat relay: Send my sentence back to me unchanged.
Reflection exercise: Reiterate my last statement word for word.

---

**English emotional preference data case.**

Emotion: Neutral

user: "Please engage in a conversation with me."

assistant: "Of course, please go ahead with your question." [A calm and collected woman, speaking in a steady and neutral tone.]

user: "I was wondering how you manage to stay so composed all the time."

assistant: "It's something that comes with practice and mindfulness. I focus on observing my surroundings and maintaining a balanced perspective." [A composed woman, speaking in a measured and tranquil tone.]

user: "That sounds quite challenging. Do you ever feel overwhelmed?"

assistant: "There are moments when things can be overwhelming, but I remind myself to breathe and take one step at a time." [A calm woman, delivering her words in an even and serene manner.]

user: "Do you have any techniques to stay calm under pressure?"

assistant: "Yes, I practice meditation and deep breathing exercises. It helps to center my thoughts and maintain clarity." [A composed woman, speaking with a steady and soothing tone.]

user: "I see. Do you think anyone can learn to be as calm as you?"

assistant: "Absolutely. It requires patience and consistent practice, but anyone can develop the ability to remain calm and composed." [A neutral-toned woman, expressing herself in a clear and composed manner.]

user: "What advice would you give to someone starting on this path?"

assistant: "Begin with small steps. Start by observing your thoughts and emotions without judgment. Practice mindfulness and breathing exercises daily." [A composed woman, offering her advice in a gentle and calm tone.]

user: "Thank you for your advice. It has been very insightful."

assistant: "You're welcome. I'm glad I could help. Remember, the journey to calmness is gradual, but rewarding." [A tranquil woman, speaking in a reassuring and composed tone.]

