# OpenReview forum: "OpenOmni: Advancing Open-Source Omnimodal Large Language Models with Progressive Multimodal Alignment and Real-time Emotional Speech Synthesis"
_NeurIPS.cc/2025/Conference — NeurIPS 2025 poster_

### Official Review · Reviewer_ULJw · 2025-06-26

**Clarity:** 3
**Significance:** 3
**Originality:** 2
**Rating:** 4
**Confidence:** 3

**Summary:**

The paper presents technical details of various stages in developing “OpenOmni” – an omnimodal LLM (vision, speech, text). The development includes two main parts: (i) omnimodal alignment and (ii) speech generation.

First, in omnimodal alignment, the process involves: (Section3.1) speech-text alignment where speech encoder / adapter is trained on speech-text data (similar to Qwen-Audio, SALMONN, etc); (Section3.2) visual-text alignment including visual encoder pretraining and visual encoder + LLM instruction tuning similar to previous works on visual LLMs (e.g., LlaVA).

Second, in speech generation, the process involves: (i) speech decoder converting LLM output representation into discrete units (output representation –[upscale + transformer layers]--> discrete unit –[vocoder]--> audio) similar to the architecture of Llama-Omni (with an exception of the MoE component); (ii) emotion speech generation via DPO, similar to Emo-DPO.

The experimental section provides implementation recipes to develop such as a model, including data in various stages. Evaluation is conducted on OmniBench, and compared against existing omnimodal models.

Key results are
- Omni and Vision performance: Compared to VITA, OpenOmni achieves a higher score on OmniBench without using tri-modal data.
- Speech performance: It performs better than VITA, while being comparable to open-source speech LLMs. Evaluation of the emotion of generated speech shows slight improvements in both English and Chinese.

Overall, this paper provides a technical report on developing OpenOmni. The paper consists of implementation details and experimental results, which can be useful for the open-source LLM development community.

**Questions:**

First of all, it should be noted that I like this paper as a technical report of a model development, and it could be useful for the open-source community, focusing on multimodal LLMs. However, could you highlight the key innovations, novelties, or findings that were learned during the development of OpenOmni?

**Ethical Concerns:**

["NO or VERY MINOR ethics concerns only"]

**Final Justification:**

I have read the rebuttal, and with further details and ablations, I believe this work could benefit the open-source community in building omnimodal LLMs. I've raised my judgment accordingly. However, in terms of its technical novelty, several points have been proposed and used by prior models.

**Limitations:**

yes

**Quality:**

3

**Strengths And Weaknesses:**

**Strengths**
- The paper provides technical details on developing OpenOmni with details on architectures, training stages, data, and evaluation results.
- Empirical results reveal that it can achieve better performance when using multi-stage omni alignment, without tri-modal data (i.e., when compared to VITA).
- The implementation details and experimental results can be useful for the open-source LLM development community.

**Weaknesses**
There is little new novelty in this paper. In my opinion, this paper reads more like a technical report of a model development. It adopts existing components or ideas, and stitches them together into a model.

The paper would benefit from a deep analysis into ablations of each component (e.g., some questions to investigate can be: how much data does one need to align omnimodality?, Does it learn cross-lingual skills (e.g., visual-text only in English, but speech-text in many languages)?, What are the weaknesses of OpenOmni compared to VITA or SOTA proprietary models such as GPT-4o, or even newer open-source like Qwen2.5-Omni, Is the current emotion speech (through DPO) good enough compared to proprietary and what might be other options to enhance the emotion speech generation further?

---

> ### Author Rebuttal · Authors · 2025-07-31
>
> Thanks for your professional and careful review. We respond to your concerns or questions as follows.
>
> > **W1&Q5**: There is little new novelty in this paper. In my opinion, this paper reads.... However, could you highlight the key innovations, novelties, or findings ..
>
> **Response:**
>
> We sincerely thank the reviewer for their thoughtful feedback. While many multimodal learning works may seem to simply integrate pretrained components and train them jointly, we believe their **novelty lies in addressing specific challenges and developing solutions**, which ultimately shape the uniqueness of the work. OpenOmni, as an open-source solution, aims to enable efficient training and competitive performance under limited data and computational resources, requiring just **40GB peak GPU memory**. By making omnimodal learning accessible to the community, OpenOmni bridges the gap between academic research and proprietary models.
>
> ##### **Key Innovations, Novelties, and Findings**
>
> 1. **Dynamic Progressive Alignment Strategy**:
>    OpenOmni uses a **progressive alignment strategy** to optimize computational resources and data use, achieving **shallow alignment** for text-speech and **deep alignment** for image-text tasks. Ablation studies in **Tables 7 and 8** validate its ability to improve both efficiency and performance compared to joint training.
> 2. **Robust Speech Decoding with TGM Module**:
>    The **TGM module** stabilizes training for the **CTC-based speech decoder**, enabling real-time multilingual speech synthesis while enhancing quality comparable to autoregressive models like **EMOVA** and **VITA**. **Figure 4** confirms its effectiveness.
> 3. **Multilingual Support with MOE Module**:
>    Conventional CTC-based methods (e.g., **LLaMA-Omni**) are limited to single-language speech generation. OpenOmni introduces the **MOE module**, supporting effective multilingual synthesis during CTC training.
> 4. **CTC-DPO Algorithm for Emotional Speech**:
>    OpenOmni’s **CTC-DPO algorithm** generates contextually consistent emotional speech for multi-turn dialogues by leveraging multi-turn datasets, bridging the user experience gap with proprietary models for adaptive interaction.
> 5. **Impact on Subsequent Research**:
>    OpenOmni’s innovations have advanced later works:
>    - **TGM module** improves speech decoder performance [1,2].
>    - Multi-turn emotional consistency enhances multi-character interactions [3].
>
> These contributions underscore OpenOmni’s progress in omnimodal learning, lowering research barriers and fostering broader community participation to advance the field collaboratively.
>
> > **Q1**: how much data does one need to align omnimodality?
>
> **Response:**
>
> Thank you for the reviewer’s insightful feedback. Appendix Tables 7 and 8 present preliminary findings from our ablation experiments on the omnimodal alignment strategy:
>
> 1. **Alignment Order**: No significant performance difference exists between **text-speech first** and **image-text first** alignments, as speech and image align independently to the text space.
>
> 2. **Alignment Depth**: **Shallow alignment** (freezing the LLM) is effective for text-speech due to their natural temporal correlation, while **deep alignment** (unfreezing the LLM) is required for image-text to achieve better results.
>
> We hypothesize that text-speech alignment demands far less data than image-text alignment. To verify this, we conducted experiments using varying training data amounts, as detailed below.
>
> | Speech      | AI-Shell2-Dev (CER)$\downarrow$ | LibriSpeech-Test-clean (WER)$\downarrow$ |
> | ----------- | ------------------------------- | ---------------------------------------- |
> | 100 Hours   | 16.2                            | 14.7                                     |
> | 500 Hours   | 8.6                             | 6.9                                      |
> | 900 Hours   | 7.2                             | 4.0                                      |
> | 1,300 Hours | 6.9                             | 3.3                                      |
> | 1,600 Hours | 6.8                             | 3.1                                      |
>
> | Image | MMStar | MMB  | MMMU | AI2D |
> | ----- | ------ | ---- | ---- | ---- |
> | 0.2M  | 35.6   | 46.3 | 29.5 | 54.6 |
> | 0.6M  | 43.5   | 60.8 | 37.9 | 64.0 |
> | 1.0M  | 49.8   | 69.9 | 42.3 | 70.4 |
> | 1.4M  | 51.2   | 72.5 | 44.1 | 72.3 |
> | 1.7M  | 52.3   | 76.2 | 46.7 | 74.8 |
>
> For text-speech alignment, **900 hours of data** achieved **97%** of the performance obtained with **1,600 hours**, showing the efficiency of natural temporal correspondence where **data quality** outweighs **quantity** in later stages.
>
> For image-text alignment, performance improves as data increases, with slower gains in later stages but room for growth, suggesting a larger gap between image and text representations where additional data enhances alignment.
>
> These findings highlight differing data requirements for text-speech and image-text alignments, offering valuable insights for future research.
>
> > **Q2**: Does it learn cross-lingual skills?
>
> **Response:**
>
> To evaluate multilingual generalization during speech-text alignment, we used the **Multilingual Speech Commands Dataset** validation set, containing speech data in 14 non-English languages. Zero-shot command recognition experiments were conducted on OpenOmni during text-speech alignment, and the accuracy results are shown in the table below.
>
> | Steps     | Kazakh      | Russian     | Tatar       | Arabic      | Turkish    | French    | German          |
> | --------- | ----------- | ----------- | ----------- | ----------- | ---------- | --------- | --------------- |
> | 100       | 12.8        | 38.5        | 20.3        | 25.5        | 22.7       | 30.1      | 34.2            |
> | 300       | 18.7        | 43.8        | 27.1        | 31.0        | 28.3       | 35.8      | 40.9            |
> | 500       | 25.2        | 48.9        | 34.0        | 37.0        | 34.2       | 41.5      | 47.6            |
> | **Steps** | **Spanish** | **Italian** | **Catalan** | **Persian** | **Polish** | **Dutch** | **Kinyarwanda** |
> | 100       | 30.2        | 28.1        | 22.5        | 18.3        | 20.4       | 24.7      | 10.8            |
> | 300       | 36.0        | 33.9        | 29.2        | 23.7        | 26.0       | 30.6      | 16.4            |
> | 500       | 41.8        | 39.7        | 35.8        | 29.1        | 31.5       | 36.4      | 22.0            |
>
> As shown in the table, OpenOmni’s text-speech alignment with Chinese and English data demonstrates strong cross-lingual generalization, enabling consistent improvements in non-English speech recognition.
>
> > **Q3**: What are the weaknesses of OpenOmni compared to ..,
>
> **Response:**
>
> We appreciate the reviewer’s detailed feedback and would like to clarify the distinctions between **OpenOmni**, **VITA**, and proprietary models like **GPT-4o**:
>
> 1. **Architectural Differences**:  VITA uses a pipeline approach, generating text responses first and then passing them to a TTS model, which risks **error accumulation**. It also requires duplex communication with two model deployments, significantly increasing GPU memory needs. OpenOmni adopts an **end-to-end design**, feeding LLM features into a **CTC-based decoder** for real-time parallel decoding, enabling **emotionally consistent speech** in multi-turn dialogues—something VITA struggles with.
>
> 2. **Training Strategies**:  OpenOmni employs a **progressive alignment strategy** that reduces tri-modal data dependency and increases efficiency, requiring only **40GB peak GPU memory** and **1,400 GPU Hours** for strong performance. By contrast, VITA uses **joint training**, consuming **three times more data** (**5M vs. 1.7M**) and **80GB GPU memory**, with nearly **10,000 GPU Hours**, making it significantly more computationally expensive.
>
> 3. **Advantages of OpenOmni**:  OpenOmni is **faster, more efficient**, and excels at producing **emotionally consistent speech** for better interaction experiences. Its design lowers research and deployment barriers, making omnimodal research more accessible.
>
> 4. **Comparison with Proprietary Models**:  OpenOmni lacks access to large-scale industry datasets or computational resources like **GPT-4o** and **Qwen2.5-Omni**. While its **CTC-based decoder** offers efficient real-time emotional speech generation, speech quality is slightly inferior to the autoregressive methods used in proprietary models.
>
> In summary, OpenOmni optimizes efficiency, accessibility, and performance, contributing significantly to omnimodal research while lowering entry barriers for the academic community.
>
> > **Q4**: Is ... emotion speech (through DPO) good enough .. what might be other options to ..
>
> **Response:**
>
> Thank you for the reviewer’s feedback. We are pleased to discuss emotion speech generation further.
>
> Although OpenOmni still has gaps compared to proprietary models like **GPT-4o**, its focus on **multi-turn emotional speech generation** and strong context-learning capabilities has significantly narrowed the gaps with previous methods (such as **EMOVA** and **Qwen-Audio**) and proprietary models.
>
> Recent studies [4,5] show that large-scale datasets of millions of hours enable adaptive, contextually coherent speech generation. However, such data and resources remain beyond the academic community’s reach. Meanwhile, **post-training strategies like CTC-DPO in OpenOmni** provide a practical, efficient solution for producing reliably consistent emotional speech.
>
>
>
> [1] Stream-Omni: Simultaneous Multimodal Interactions with Large Language-Vision-Speech Model
>
> [2] LLaMA-Omni 2: LLM-based Real-time Spoken Chatbot with Autoregressive Streaming Speech Synthesis
>
> [3] OmniCharacter: Towards Immersive Role-Playing Agents with Seamless Speech-Language Personality Interaction
>
> [4] Higgs Audio V2: Redefining Expressiveness in Audio Generation
>
> [5] MOSS: Text to Spoken Dialogue Generation

---

> > ### Author Response · Authors · 2025-08-04
> >
> > Dear Reviewer ULJw:
> >
> > Thank you immensely for your valuable feedback on our manuscript. We've worked diligently to incorporate your suggestions and make necessary revisions. With the review timeline approaching, we kindly ask if you could spare a moment to re-evaluate the updated document. Please let us know if there is anything else you need from our end for clarification.
> >
> > We truly appreciate your cooperation and continued support.
> >
> > Warm regards,
> >
> > Authors

---

> > ### Comment · Reviewer_ULJw · 2025-08-04
> >
> > Thank you for the detailed rebuttal. I appreciate the effort to clarify the novelty and provide more details on ablations & comparisons. Although I still view various parts of work as engineering efforts (e.g., training strategies and using multiple languages, etc), I'm willing to and I've raised my judgment as this effort with clear details could help the open source community.

---

> > > ### Author Response · Authors · 2025-08-05
> > >
> > > Thank you so much for taking the time to share your feedback! We're glad to hear that your concerns have been addressed. Your support means a lot to us, and we truly appreciate your willingness to improve your rating score. If you have other comments, we are happy to address them to polish this work. We look forward to contributing to the development of both the omni-modal research and the open-source community.

---

### Official Review · Reviewer_GJPB · 2025-07-01

**Clarity:** 3
**Significance:** 2
**Originality:** 2
**Rating:** 4
**Confidence:** 4

**Summary:**

This paper introduces OpenOmni, an open-source omni-modal large language model designed to unify vision, text, and speech modalities via a progressive two-stage training pipeline. First, OpenOmni aligns speech-text and image-text data independently using pretrained encoders and language models. Then, it learns a lightweight streaming speech decoder for real-time and emotionally expressive speech synthesis, using DPO. The authors propose that this decomposed strategy bypasses the need for tri-modal datasets while achieving better generalization.

**Questions:**

1.	How exactly does OpenOmni’s alignment strategy differ from that in Qwen-Omni or AnyGPT, which also used text-pivot alignment and pre-trained encoders? A quantitative or ablation-based comparison would strengthen the contribution.

2.	The paper claims DPO is more stable than RLHF for emotional speech but does not present empirical comparisons. An ablation with RLHF or reward-based preference methods would make this claim credible.

3.	Although WER and emotion classification are used, human ratings (e.g., MOS or empathy scores) for speech naturalness and emotional congruence would strengthen the case for "self-aware" interaction. Have human evaluations been conducted?

**Ethical Concerns:**

["NO or VERY MINOR ethics concerns only"]

**Final Justification:**

After reviewing the authors’ rebuttal and the discussion among reviewers, I find that several of my earlier concerns, particularly regarding the clarity of contributions and differences from prior work, have been addressed through additional explanations, comparisons, and ablation results. Given that the reasons to accept now slightly outweigh the reasons to reject, I am updating my rating to borderline accept.

**Limitations:**

yes

**Quality:**

3

**Strengths And Weaknesses:**

Strengths:

1.	The decomposition into two separate alignment stages (speech-text and image-text) using text as a pivot is a pragmatic workaround for the scarcity of high-quality tri-modal datasets.

2.	The integration of a non-autoregressive speech decoder trained with DPO for emotional nuance is an engineering contribution. It improves latency and user experience in real-time multimodal dialogue systems.

Weaknesses:
1.	The proposed progressive alignment (text as pivot) is not novel. The approach is reminiscent of modality bridging used in prior work like AnyGPT and Qwen-Omni, both of which are cited but not sufficiently contrasted with in terms of technical differences.

2.	OpenOmni largely reuses existing components (Qwen2.5 LLM, CLIP, Whisper, GLM4-Voice tokenizer) and standard training techniques (CTC, MoE, instruction tuning). The speech decoder architecture is mostly standard with little new insight.

3.	The descriptions are exhaustive but often lack clarity about what exactly is novel or generalizable. For example, the TGM module’s design is described but not evaluated independently, and its impact remains unclear.

4.	The progressive alignment strategy, DPO training for emotional speech, and use of speech decoders are all incremental rather than truly novel. Several prior works cited by the authors (e.g., Qwen2-Audio, EMOVA, AnyGPT) have either done similar decompositions or employed similar optimization strategies.

---

> ### Author Rebuttal · Authors · 2025-07-31
>
> Thanks for your professional and careful review. We respond to your concerns or questions as follows.
>
> > **W1**&**Q1**: progressive alignment (text as pivot) is not novel. The approach...technical differences. How exactly ...? A quantitative or ablation-based comparison...
>
> **Response:**
>
> We sincerely appreciate the reviewer’s feedback. While most multimodal learning methods [1,2,3] commonly use **text as a pivot** as a foundational approach, their **novelty and innovation** often lie in the **specific alignment strategies** tailored to applications. To highlight OpenOmni’s contributions, we compare its alignment strategy with previous works such as **AnyGPT** and **Qwen-Omni** and summarize the differences in the table below.
>
> | Method   | Data  | Minimum  Memory | joint training | tri-modal data | freeze LLM (speech) | freeze LLM (image) |
> | -------- | ----- | --------------- | -------------- | -------------- | ------------------- | ------------------ |
> | Any-GPT  | ~10M  | 60G             | $\checkmark$   | $\checkmark$   | $\checkmark$        | $\checkmark$       |
> | OpenOmni | 1.7M  | 40G             | $\times$       | $\times$       | $\checkmark$        | $\times$           |
> | QwenOmni | ~500M | 100G            | $\checkmark$   | $\checkmark$   | $\times$            | $\times$           |
>
> As shown in the table, **AnyGPT** and **Qwen-Omni** rely on tri-modal data and single-stage joint training for all modalities. In contrast, **OpenOmni** employs a **progressive alignment strategy**, aligning modalities in separate stages. Each stage involves at most **two-modality alignment**, reducing GPU memory requirements and lowering the entry barrier for omnimodal learning.
>
> This approach allows flexible training strategies. Ablation studies in **Appendix Tables 7 and 8** show that **text-speech alignment** achieves competitive results by freezing the LLM, improving training efficiency by **300%**, while **image-text alignment** requires unfreezing the LLM due to larger modality gaps for deeper alignment.
>
> By tailoring strategies to specific modality characteristics, OpenOmni’s **dynamic training strategy** enhances flexibility, improves efficiency, and maintains competitive results. With less data, OpenOmni achieves **SOTA performance** compared to other open-source models, showcasing its effectiveness and innovation in omnimodal learning.
>
> > **W2**: OpenOmni largely reuses existing components .... The speech decoder architecture is mostly ....
>
> **Response:**
>
> Thank you for the reviewer’s valuable feedback. While the reuse of existing components and standard training techniques is common across omnimodal learning frameworks, **OpenOmni** sets itself apart through its **dynamic and flexible modal progressive alignment strategy**, which minimizes dependencies on tri-modal data and GPU requirements—marking a key innovation compared to previous works.
>
> Furthermore, OpenOmni utilizes a lightweight **CTC-based speech decoder** with notable improvements. It introduces the **MOE mechanism** to handle multilingual conflicts during CTC training and the **TGM module**, which stabilizes the CTC training process. These enhancements retain the real-time efficiency of CTC decoders while achieving speech generation quality on par with autoregressive methods.
>
> These contributions highlight OpenOmni's **innovations and findings**, offering valuable insights to the omnimodal learning community and driving advancements in this field.
>
> > **W3**: For example, the TGM module’s design is described but ...
>
> **Response:**
>
> Thank you for the reviewer’s feedback. In Appendix Lines 650-654, we qualitatively analyzed Table 4, showing the **TGM module** significantly stabilizes training for both **CTC** and **AR mode decoders**, enhancing speech generation. Additionally, we supplemented this with quantitative experiments for a more comprehensive evaluation.
>
> | mode               | AI-Shell2-Dev (CER)$\downarrow$ | LibriSpeech-Test-clean (WER)$\downarrow$ |
> | ------------------ | ------------------------------- | ---------------------------------------- |
> | AR w/o TGM         | 7.4                             | 3.5                                      |
> | AR w/ TGM          | 6.9                             | 2.3                                      |
> | NAR (CTC)  w/o TGM | 8.7                             | 4.3                                      |
> | NAR (CTC) w/ TGM   | 7.3                             | 2.6                                      |
>
> As shown in the table, the **TGM module** significantly improves content consistency in both **AR** and **CTC modes**, with strong generalization for Chinese and English speech generation. It has also been adopted by subsequent works [4,5] to enhance speech decoder consistency.
>
> > **W4**: The progressive alignment strategy...rather than truly novel. Several prior works (e.g., Qwen2-Audio, EMOVA, AnyGPT) have either done similar ....
>
> **Response:**
>
> Thank you for the reviewer’s valuable comments. We clarify the differences between our method and previous works in terms of **progressive alignment strategy**, **DPO training for emotional speech**, and **speech decoders**:
>
> 1. **Progressive Alignment Strategy**: OpenOmni employs a flexible modal alignment approach based on ablation experiments, using different strategies for **text-speech** and **image-text alignments**. This achieves competitive results with minimal computational resources and training data.
> 2. **Speech Decoders**: OpenOmni uses a **CTC-based approach** for parallel real-time speech generation, unlike autoregressive methods in **AnyGPT** and **EMOVA**. We enhance CTC to support bilingual speech and yield results comparable to autoregressive techniques while maintaining efficiency.
> 3. **DPO Training**: OpenOmni focuses on emotional consistency in multi-turn dialogues, similar to **GPT-4o**, allowing the model to learn user intent through context and output appropriate emotional speech for better interaction. This approach has been adopted by subsequent work [6] to improve multi-turn dialogue experiences. In contrast, methods like **EMOVA** rely on predefined emotion keywords for single-turn tasks, limiting their effectiveness in multi-turn scenarios.
>
> From these perspectives, OpenOmni fundamentally differs from previous methods.
>
> > **Q2**:The paper claims DPO is more stable.... An ablation with RLHF or reward-based ...
>
> **Response:**
>
> Thank you for the reviewer’s comments. We conducted ablation experiments using the PPO algorithm and Verl framework, with the **EO2S-9K** dataset to train the reward model for RLHF. The experimental results and computational resource comparisons are shown in the table below.
>
> | GPU*Hour | Method | Lang | Angry & Disgusted | Fearful  | Happy    | Neutral  | Other    | Sad      | Surprised |
> | -------- | ------ | ---- | ----------------- | -------- | -------- | -------- | -------- | -------- | --------- |
> | 54       | DPO    | ZH   | **96.6**          | 78.4     | **37.7** | **97.1** | 62.8     | 90.7     | 29.8      |
> | 130      | PPO    | ZH   | 94.7              | **79.2** | 36.1     | 94.5     | **64.3** | **91.6** | **30.9**  |
> | 54       | DPO    | EN   | 91.3              | **70.4** | **60.6** | **94.6** | 49.6     | 77.3     | **13.9**  |
> | 130      | PPO    | EN   | **91.6**          | 69.4     | 59.8     | 92.7     | **50.3** | **79.5** | 10.6      |
>
> As shown in the table, while PPO shows no significant performance advantage over DPO, it requires **loading and training four model copies**, resulting in much higher GPU resource consumption.
>
> Moreover, during training, the **EO2S-9K** reward model often failed to provide accurate evaluations, causing instability in PPO. Outlier data, where the policy model generated unusual speech and the reward model delivered incorrect emotional judgments, repeatedly led to training failures, requiring **three attempts** to complete.
>
> In contrast, DPO is unaffected by reward model performance on small datasets, offering more stable training, lower resource consumption, and results comparable to PPO, making it a more efficient and practical choice.
>
> > **Q3**: Although WER and emotion classification .... Have human evaluations been conducted?
>
> **Response:**
>
> Thank you for the reviewer’s comments. We used the **MOS (Mean Opinion Score)** metric, with **10 experts** evaluating the generated speech for Chinese and English sets and EO2S emotional speech. The experimental results are shown in the table below.
>
> | mode         | AI-Shell2-Dev (MOS)$\uparrow$ | LibriSpeech-Test-clean (MOS)$\uparrow$ | EO2S (MOS)$\uparrow$ |
> | ------------ | ----------------------------- | -------------------------------------- | -------------------- |
> | ground_truth | 4.50                          | 4.50                                   | 4.50                 |
> | AR           | 3.89                          | 4.22                                   | 4.14                 |
> | NAR (CTC)    | 3.83                          | 4.16                                   | 4.08                 |
>
> The experimental results indicate that the MOS evaluations from human experts demonstrate that the speech generated by OpenOmni is **natural, fluent, and exhibits emotional consistency**.
>
> [1] LLaVA: Large Language and Vision Assistant
>
> [2] AnyGPT: Unified Multimodal LLM with Discrete Sequence Modeling
>
> [3] EMOVA: Empowering Language Models to See, Hear and Speak with Vivid Emotions
>
> [4] Stream-Omni: Simultaneous Multimodal Interactions with Large Language-Vision-Speech Model
>
> [5] LLaMA-Omni 2: LLM-based Real-time Spoken Chatbot with Autoregressive Streaming Speech Synthesis
>
> [6] OmniCharacter: Towards Immersive Role-Playing Agents with Seamless Speech-Language Personality Interaction

---

> > ### Author Response · Authors · 2025-08-04
> >
> > Dear Reviewer GJPB:
> >
> > We are truly grateful for your insightful comments and the guidance you provided during your review of our paper. We are pleased to inform you that we have addressed all points raised and have made significant improvements. As the discussion phase draws near, we kindly request your reevaluation at your earliest convenience. Should any questions remain, we are at your disposal to clarify them promptly.
> >
> > Thank you for your time and understanding.
> >
> > Sincerely,
> >
> > Authors

---

> > > ### Author Response · Authors · 2025-08-07
> > >
> > > Dear Reviewer GJPB:
> > >
> > > Thank you immensely for your valuable feedback on our manuscript. We've worked diligently to incorporate your suggestions and make necessary revisions. With the review timeline approaching, we kindly ask if you could spare a moment to re-evaluate the updated document. Please let us know if there is anything else you need from our end for clarification.
> > >
> > > We truly appreciate your cooperation and continued support.
> > >
> > > Warm regards,
> > >
> > > Authors

---

> ### Author Response · Authors · 2025-08-08
>
> Dear Reviewer GJPB,
>
> I hope this message finds you well. As the discussion period is nearing its end with only one day remaining, I wanted to ensure we have addressed all your concerns satisfactorily. We have provided comprehensive responses to all four weaknesses and three questions you raised, including the quantitative comparisons and human evaluations you requested.
>
> If there are any additional points or feedback you'd like us to consider, please let us know. Your insights are invaluable to us, and we're eager to address any remaining issues to improve our work.
>
> Thank you for your time and effort in reviewing our paper.
>
> Sincerely,
>
> Author

---

### Official Review · Reviewer_TCBG · 2025-07-03

**Clarity:** 3
**Significance:** 3
**Originality:** 3
**Rating:** 5
**Confidence:** 3

**Summary:**

This paper introduces a novel two-stage training framework to address key limitations in open-source omnimodal large language models, specifically the scarcity of high-quality tri-modal datasets and challenges in real-time emotional speech synthesis. The framework first aligns speech-text and image-text modalities independently using a progressive alignment strategy, enabling near-zero-shot generalization from vision to speech without tri-modal data. It then trains a lightweight non-autoregressive speech decoder optimized via DPO for real-time, emotionally coherent speech generation. Evaluations across omnimodal, vision-language, and speech-language benchmarks show that OpenOmni outperforms SOTA models like VITA while using 5× fewer training examples and a smaller model size. It achieves sub-second latency and a 7.7% improvement in emotion classification accuracy.

**Questions:**

The two-stage training procedure is well elaborated in the paper and the appendix. However, I'm still curious how the authors dealt with catastrophic forgetting?

**Ethical Concerns:**

["NO or VERY MINOR ethics concerns only"]

**Final Justification:**

The authors have provided supplementary experiments and explanations regarding the choice of CTC, and I can understand that they chose CTC for lower latency and less computational load. However, I would prefer to optimize AR inference using a mature inference framework rather than choosing CTC, which has poorer performance. The authors have further elaborated on the phenomenon that text&symbols and count&quantity have poorer performance, and also provided a detailed explanation of the training process. I think the overall paper is interesting, despite some unclear writing.

**Limitations:**

yes.

**Paper Formatting Concerns:**

No major formatting issues.

**Quality:**

3

**Strengths And Weaknesses:**

**Strengths**
1. The authors proposed a smaller model with fewer training samples to achieve SOTA performance in omnimodal learning tasks, which is of great value to the academic community.
2. The authors found a quasi-zero-shot transfer capability in the proposed model, which is insightful and has the value of further research.
3. The paper is well organized, and the design choices are elaborated in detail with thorough verification.

**Weaknesses**
1. The authors adopted CTC loss to train the speech decoder instead of the next-token-prediction paradigm, which is more commonly used, such as Kimi-Audio [1] and Moshi [2]. Please clarify the decision.
2. The authors introduced the training procedures as two stages (line 51), but the elaborations indicate three stages (Figure 2, Appendix A). I understand that the authors intended to show that the omnimodal learning capability is trained with the first two stages, and the third stage is for speech generation. However, the clarification needs improving.
3. Table 1 shows that the proposed model underperforms significantly in terms of text&symbols, count&quantity. Please analyze and discuss the reasons.

[1] Kimi-Audio Technical Report. https://arxiv.org/abs/2504.18425.
[2] Moshi: a speech-text foundation model for real-time dialogue. https://arxiv.org/abs/2410.00037

---

> ### Author Rebuttal · Authors · 2025-07-31
>
> Thanks for your professional and careful review. We respond to your concerns or questions as follows.
>
> > **W1**: The authors adopted CTC loss to train the speech decoder instead of the next-token-prediction paradigm, which is more commonly used, such as Kimi-Audio [1] and Moshi [2]. Please clarify the decision.
>
> **Response:**
>
> Thank you for the reviewer’s detailed comments. To further explain our choice of using the CTC decoding approach, we conducted the following experiments for comparison.
>
> In addition to the basic metrics for semantic consistency of speech content, such as WER (Word Error Rate) and CER (Character Error Rate), we also employed 10 expert evaluators to assess the MOS (Mean Opinion Score), which evaluates speech naturalness and emotional congruence.
>
> These experiments provide a comprehensive evaluation framework to highlight the advantages of the CTC decoding approach.
>
> | mode         | Speed | GPU Memory | AI-Shell2-Dev (CER)$\downarrow$ | AI-Shell2-Dev (MOS)$\uparrow$ | LibriSpeech-Test-clean (WER)$\downarrow$ | LibriSpeech-Test-clean (MOS)$\uparrow$ |
> | ------------ | ----- | ---------- | ------------------------------- | ----------------------------- | ---------------------------------------- | -------------------------------------- |
> | ground_truth |       |            | 3.4                             | 4.50                          | 1.6                                      | 4.50                                   |
> | AR           | 1.8s  | 18654MB    | 6.9                             | 3.89                          | 2.3                                      | 4.22                                   |
> | NAR (CTC)    | 226ms | 12786MB    | 7.3                             | 3.83                          | 2.6                                      | 4.16                                   |
>
> As shown in the table, the CTC speech decoder enhanced by the TGM module maintains real-time generation capabilities while using less GPU memory. Compared to autoregressive speech decoders, it produces competitive, fluent, and natural speech with high MOS scores evaluated by human experts. This supports our decision to adopt the CTC speech decoder to improve real-time interaction experiences.
>
> > **W2**: the clarification needs improving.
>
> **Response:**
>
> Thank you for the reviewer’s careful and thoughtful feedback, which has helped improve the readability of our work. We intend to clarify that the omnimodal learning capability is developed through the first two stages, while the third stage is dedicated to speech generation. These clarifications will significantly enhance the clarity of our paper, and we are committed to implementing these changes thoughtfully in the revised version. Thank you again for your constructive suggestions.
>
> > **W3**: Table 1 shows that the proposed model underperforms significantly in terms of text&symbols, count&quantity. Please analyze and discuss the reasons.
>
> **Response:**
>
> Thank you for the reviewer’s careful feedback. We hypothesize that this performance discrepancy might stem from OpenOmni’s reliance on the image data distribution of **MMEvol**, which contains relatively few samples for **text & symbol** and **count & quantity** tasks.
>
> To validate this hypothesis, we sampled 50K text & symbol data from the Video-SALMONN training dataset and 50K count & quantity data from the Pixmo-Count dataset. These samples were mixed with the original 1.7M instruction data, forming a 1.8M instruction dataset for use in OpenOmni's second-stage alignment training. The experimental results are reported below.
>
> | Data | Action Activity | Story Description | Plot Inference | Identification Description | Contextual  Environmental | Identity Relationship | Text  Symbols | Count  Quantity |
> | ---- | --------------- | ----------------- | -------------- | -------------------------- | ------------------------- | --------------------- | ------------- | --------------- |
> | 1.7M | 36.65           | 45.65             | 32.91          | 44.08                      | 48.23                     | 34.38                 | 24.00         | 33.33           |
> | 1.8M | 37.12           | 45.82             | 33.63          | 44.32                      | 48.47                     | 34.85                 | **49.64**     | **50.89**       |
>
> As shown in the table, after incorporating the additional training data specifically targeting text & symbol and count & quantity, OpenOmni demonstrates significant performance improvements in these two capabilities.
>
> > **Q1**: The two-stage training procedure is well elaborated in the paper and the appendix. However, I'm still curious how the authors dealt with catastrophic forgetting?
>
> **Response:**
>
> Thank you for the reviewer’s valuable comments. We are happy to further elaborate on our strategies for addressing catastrophic forgetting during the multi-stage progressive omnimodal learning process.
>
> During the text-speech alignment stage, we adopt a **shallow alignment strategy** by freezing the LLM during training. This approach ensures that catastrophic forgetting does not occur.
>
> In contrast, during the image-text alignment stage, we adopt a **deep alignment strategy** by unfreezing the LLM for training. While this improves image-text alignment capabilities, it can influence the original knowledge of the LLM and potentially disrupt the text-speech alignment knowledge established in the first stage, leading to catastrophic forgetting. To mitigate catastrophic forgetting for the LLM, we employ a mature approach inspired by MMEvol and LLaVA, mixing 143K high-quality textual instruction data from WizardLM into the training process. This strategy effectively alleviates catastrophic forgetting in the LLM.
>
> As demonstrated in our ablation study in Appendix Table 7, the impact of the image-text alignment stage on the knowledge from the text-speech alignment stage is minimal. Specifically, unfreezing the LLM for training in the second stage causes only a 2.8% performance drop (12.10 vs. 12.45) on AI-Shell and a 2.1% performance drop (11.55 vs. 11.8) on LibriSpeech, with no catastrophic forgetting observed.
>
> During the speech generation stage, the LLM remains frozen, ensuring that catastrophic forgetting does not occur at this stage either.
>
> We hope this clarification highlights the robustness of OpenOmni’s multi-stage learning process and strategies for effectively handling catastrophic forgetting.

---

> > ### Author Response · Authors · 2025-08-04
> >
> > Dear Reviewer TCBG,
> >
> > We hope this message finds you well. We are writing to sincerely thank you for taking the time to review our rebuttal and for providing such insightful and constructive feedback on our submission. Your expertise and attention to detail are truly appreciated and have been incredibly beneficial in guiding our research.
> >
> > The suggestions you provided have opened up new perspectives and opportunities for enhancing the quality of our work. We are committed to implementing your recommendations to the best of our ability and are eager to explore the ideas you have mentioned further.
> >
> > Thank you once again for your invaluable contribution to our research. Your support and guidance are deeply appreciated, and we are hopeful that the revisions will meet your expectations.
> >
> > Sincerely,
> >
> > Author

---

> > ### Comment · Reviewer_TCBG · 2025-08-08
> > **Official Comment by Reviewer TCBG**
> >
> > I appreciate the authors' detailed responses. The authors have provided supplementary experiments and explanations regarding the choice of CTC, and I can understand that they chose CTC for lower latency and less computational load. However, I would prefer to optimize AR inference using a mature inference framework rather than choosing CTC, which has poorer performance. The authors have further elaborated on the phenomenon that text&symbols and count&quantity have poorer performance, and also provided a detailed explanation of the training process. After carefully reading the authors' rebuttal and the comments from other reviewers, I believe the authors have basically addressed most of my doubts about the paper, and I will update my rating.

---

> ### Author Response · Authors · 2025-08-07
>
> Dear Reviewer TCBG:
>
> Thank you again for your time and valuable feedback on our paper. We noticed that your official comment on OpenReview is still pending. Could you please let us know if there’s any additional information needed from our side? We’d greatly appreciate it if you could submit your comment when you have a moment.
>
> Sincerely,
>
> Authors

---

> ### Author Response · Authors · 2025-08-08
>
> Dear Reviewer TCBG,
>
> I hope this message finds you well. As the discussion period is approaching its deadline, I wanted to ensure we have adequately addressed all of your comments and concerns raised during the review.
>
> If there are any remaining issues or additional points you’d like us to clarify, please don't hesitate to let us know. Your valuable insights are critical for improving our work, and we're more than willing to engage further to address any outstanding questions.
>
> Thank you very much for your effort and time in reviewing our submission.
>
> Sincerely,
>
> Author

---

> ### Author Response · Authors · 2025-08-09
>
> Dear Reviewer TCBG,
>
> Thank you so much for taking the time to share your thoughtful feedback and for recognizing the efforts and innovations presented in our work. We are truly grateful that our responses and additional experiments have addressed most of your concerns, and we deeply appreciate your willingness to update your rating score. Your constructive feedback has significantly helped us refine and clarify our contributions, and for that, we sincerely thank you.
>
> We are especially pleased to see that you recognize the value of our dynamic progressive alignment strategy, which demonstrates how SOTA results in omnimodal learning can be achieved with minimal computational resources and a smaller data scale. This strategy showcases our focus on enabling efficient and accessible training, representing a step forward in addressing challenges such as high GPU memory demands and the need for vast amounts of data.
>
> Additionally, we’re thrilled that the capabilities of our TGM-enhanced CTC-based speech decoder stood out to you. As you observed, the lightweight and efficient design of this approach ensures that speech generation remains both competitive in quality compared to AR-based decoders and fully capable of real-time synthesis. The combination of low latency, reduced computational load, and emotionally coherent generation is pivotal for improving the user experience in interactive multimodal systems, and we’re excited about its potential to further enrich omnimodal dialogue and beyond.
>
> This balanced integration of innovation and practicality—achieving high performance under resource constraints—is a central contribution of OpenOmni that we hope continues to resonate with the omnimodal and AI research communities.
>
> If you have further suggestions or questions, we are always happy to engage in future discussions to further polish this work. We look forward to contributing to both the advancement of scientific research in this field and the strengthening of the open-source community.
>
> Thank you again for your valuable review and kind recognition. Your support inspires us to push forward.
>
> Best regards,
>
> Author

---

### Official Review · Reviewer_LJos · 2025-07-05

**Clarity:** 3
**Significance:** 3
**Originality:** 3
**Rating:** 4
**Confidence:** 4

**Summary:**

This paper addresses two key issues currently present in the community: the lack of high-quality omnimodal data and the suboptimal performance of real-time emotional speech synthesis. To tackle these challenges, the authors propose OpenOmni, a comprehensive omnimodal large Languagemodel framework equipped with Real-time Emotional Speech Synthesis capabilities. First, this paper inroduces two high-quality speech datasets—O2S-300K and EO2S-9K—to support efficient speech generation and emotional preference optimization. Then, they introduce a modality-progression training framework to achieve efficient tri-modal alignment. Experimentally, OpenOmni achieves state-of-the-art (SOTA) performance across multiple benchmarks, while also demonstrating significant improvements in training efficiency.

**Questions:**

1. In Stage 2 of the framework, which focuses on Image-Text alignment, the training involves fine-tuning the LLM. A potential concern is whether this fine-tuning could undermine the Speech-Text alignment achieved during Stage 1.

2. Table 4 shows that the improvement from DPO training on Chinese evaluation is significantly more noticeable than on English. Is this primarily due to the data distribution or the model's pre-training?

3. For the choice of speech decoder, the paper states that AR and NAR decoder differ in generation quality and speed. Please provide quantitative experimental results for both and analyze them.

**Ethical Concerns:**

["NO or VERY MINOR ethics concerns only"]

**Limitations:**

See weakness

**Quality:**

3

**Strengths And Weaknesses:**

**Strengths:**

1. High-quality omnimodal data and omnimodal alignment remain key challenges in current research on omnimodal large models. This paper tackles these two critical issues and presents effective solutions, underscoring its considerable significance to the field.

2. The proposed framework is notably efficient and streamlined. By utilizing a single large language model (LLM) and a lightweight speech decoder within a multi-stage training strategy, it delivers substantial improvements in both generation efficiency and model size, outperforming the leading open-source model, VITA.

3. The paper provides a clear and detailed description of the model architecture and training strategy.

**Weaknesses:**

1. The paper considers dataset construction as one of its contributions; however, the description of the process lacks sufficient detail and fails to clearly demonstrate the advantages of the newly constructed datasets.

2. The paper lacks comparison and discussion with other methods. For example, it does not clearly outline the improvements and differences in architecture and training strategy between OpenOmni and VITA.

---

> ### Author Rebuttal · Authors · 2025-07-31
>
> Thanks for your professional and careful review. We respond to your concerns or questions as follows.
>
> > **W1**: The paper considers dataset construction as one of its contributions; however, the description of the process lacks sufficient detail and fails to clearly demonstrate the advantages of the newly constructed datasets.
>
> **Response:**
>
> Thank you for the reviewer’s comments. In Section 4.1 of the main text, we provided a detailed explanation of our data sources and construction processes, with a brief introduction in Appendix A. Here, we further clarify and highlight the methodologies and advantages of our datasets.
>
> We sampled **300K multi-turn dialogues** from **UltralChat** (text) and **MMEvol** (multimodal), selecting longer responses to ensure extended audio durations for better model learning. Half of the data was translated into Chinese to support bilingual generation. Using **CosyVoice**, speech synthesis was performed on the text data to create **O2S-300K**.
> For **EO2S-9K**, we intensively filtered dialogues with contextual consistency and strong emotional tendencies in multi-turn interactions. Emotional speech synthesis was conducted using **CosyVoice**, resulting in a dataset optimized for DPO-based emotional speech generation. A total of **8,000 A100 GPU hours** were spent synthesizing O2S-300K and EO2S-9K, with filtering, translation, and selection costing approximately **$2,000**. These efforts ensure diverse, high-quality datasets.
>
> **Advantages**:
>
> 1. **O2S-300K**: It is the **bilingual multi-turn dataset** with tri-modal inputs (text, audio, vision). Unlike prior works like **LLaMA-Omni**, which use single-turn English-only voice data, O2S-300K enables building **multi-turn bilingual interaction models**, supporting **arbitrary input combinations** for enhanced contextual consistency and interaction experience.
>
> 2. **EO2S-9K**: It provides **fine-grained emotional speech annotations** with contextual consistency, allowing adaptive emotional speech generation after training for superior multi-turn interactions. Previous datasets rely on pre-defined emotion tags in prompts, often leading to inconsistencies between emotion tags and text (e.g., “I feel happy [sad]”), limiting adaptive learning and natural dialogue engagement.
>
> These datasets address key limitations of prior works, advancing bilingual, multi-turn, and emotionally adaptive speech synthesis. If further clarification is needed, we are happy to address additional questions.
>
> > **W2**: The paper lacks comparison and discussion with other methods. For example, it does not clearly outline the improvements and differences in architecture and training strategy between OpenOmni and VITA.
>
> **Response:**
>
> Thank you for the thoughtful feedback. We would like to clarify the key architectural and training differences between OpenOmni and VITA.
>
> 1. **Architectural Differences**:
>    - **VITA** uses a pipeline-based speech generation method, where text responses are passed to a separate TTS model. To speed up synthesis, it employs a duplex communication strategy that requires two simultaneous model deployments, doubling GPU memory usage. Its non-end-to-end design limits multi-turn dialogue context utilization, making it less effective at capturing emotional consistency.
>    - **OpenOmni** features an **end-to-end speech generation design**, where LLM outputs are directly fed into a CTC-based speech decoder for real-time parallel decoding. This approach supports context-aware emotional speech generation and ensures consistency across multi-turn interactions.
> 2. **Training Differences**:
>    - **VITA** employs **large-scale joint training**, needing **three times more tri-modal data** (5M vs. 1.7M), **80GB peak GPU memory**, and nearly **10,000 GPU Hours**, demanding significantly larger computational resources.
>    - **OpenOmni** adopts a **progressive multi-modal alignment strategy**, reducing reliance on tri-modal datasets while enhancing efficiency. It requires only **40GB peak GPU memory** and **1,400 GPU Hours**.
>
>
> In summary, OpenOmni is **faster, more efficient, and context-aware**, overcoming VITA’s limitations in emotional speech generation and resource consumption, while enabling smoother and more effective multi-turn interactions.
>
> > **Q1**: In Stage 2 of the framework, which focuses on Image-Text alignment, the training involves fine-tuning the LLM. A potential concern is whether this fine-tuning could undermine the Speech-Text alignment achieved during Stage 1.
>
> **Response:**
>
> Thank you for the reviewer’s thoughtful feedback. In fact, we have conducted relevant experiments in the ablation study presented in Table 7, and provided detailed discussions in lines 669-676 of the paper.
>
> The results demonstrate that due to the natural temporal alignment between speech and text, we choose to freeze the LLM during the speech-text alignment stage, while opting to unfreeze the LLM during the image-text alignment stage. This dynamic alignment strategy effectively balances efficiency and performance trade-offs.
>
> Additionally, we observe that unfreezing the LLM during the image-text alignment stage has only a minimal impact on the speech-text alignment knowledge from the first stage. Specifically, it results in a very minor performance decline on AI-Shell (2.8%, 12.10 vs. 12.45) and LibriSpeech (2.1%, 11.55 vs. 11.8) evaluation datasets, without causing catastrophic forgetting.
>
> By employing such a progressive alignment strategy, we achieve a nearly 300% improvement in training efficiency, making it a highly practical approach. For these reasons, we adopt this method in our proposed framework.
>
> > **Q2**: Table 4 shows that the improvement from DPO training on Chinese evaluation is significantly more noticeable than on English. Is this primarily due to the data distribution or the model's pre-training?
>
> **Response:**
>
> We believe this may be related to the varying distribution of emotion-themed dialogue data during pretraining. To investigate, we randomly sampled 10K emotional speech data from the pretraining dataset and annotated it using GPT4O-mini for emotion labeling. The distribution of different emotions in the data is shown below:
>
> | Lang | Angry & Disgusted | Fearful | Happy | Neutral | Other | Sad  | Surprised |
> | ---- | ----------------- | ------- | ----- | ------- | ----- | ---- | --------- |
> | ZH   | 0.08              | 0.10    | 0.03  | 0.63    | 0.05  | 0.09 | 0.02      |
> | EN   | 0.05              | 0.04    | 0.07  | 0.72    | 0.04  | 0.05 | 0.03      |
>
> From the table, it is evident that the differing proportions of emotions in the pretraining data contribute to the observed performance. Emotions with a higher proportion in the pretraining dataset are more likely to be effectively triggered during subsequent DPO training, thereby leading to a significant improvement in emotional speech generation capabilities.
>
> > **Q3**:For the choice of speech decoder, the paper states that AR and NAR decoder differ in generation quality and speed. Please provide quantitative experimental results for both and analyze them.
>
> **Response:**
>
> Thank you for the reviewer’s valuable comments. Here, we provide more detailed quantitative experimental results comparing AR (Autoregressive) and NAR (Non-Autoregressive) modes, along with their corresponding analysis.
>
> In addition to the baseline metrics for semantic consistency of speech content, such as WER (Word Error Rate) and CER (Character Error Rate), we also employed **10 expert evaluators** to assess the MOS (Mean Opinion Score) metric. The MOS metric provides a comprehensive evaluation of speech naturalness and emotional congruence.
>
> Through this detailed comparison, we aim to better illustrate the strengths and weaknesses of both modes and their applicability to various use cases.
>
> | mode         | Speed | GPU Memory | AI-Shell2-Dev (CER)$\downarrow$ | AI-Shell2-Dev (MOS)$\uparrow$ | LibriSpeech-Test-clean (WER)$\downarrow$ | LibriSpeech-Test-clean (MOS)$\uparrow$ |
> | ------------ | ----- | ---------- | ------------------------------- | ----------------------------- | ---------------------------------------- | -------------------------------------- |
> | ground_truth |       |            | 3.4                             | 4.50                          | 1.6                                      | 4.50                                   |
> | AR           | 1.8s  | 18654MB    | 6.9                             | 3.89                          | 2.3                                      | 4.22                                   |
> | NAR (CTC)    | 226ms | 12786MB    | 7.3                             | 3.83                          | 2.6                                      | 4.16                                   |
>
> As shown in the table, OpenOmni's NAR mode enables real-time speech synthesis with lower GPU memory usage, while still producing high-quality and fluent synthesized speech as confirmed by human evaluations.
>
> In terms of generation quality, the AR mode achieves slightly higher performance but comes at the cost of significantly greater computational requirements and GPU memory consumption. Additionally, the autoregressive nature of the AR mode limits its ability to perform real-time speech generation. In contrast, the NAR mode strikes a balance between speech generation quality and efficiency, making it a practical trade-off solution for widespread adoption.
>
> To support the open-source community and democratize OmniModal research, OpenOmni adopts the most efficient approach by integrating a progressive omnimodal alignment strategy combined with the NAR-mode speech decoder. This choice ensures that the community can participate in cutting-edge research with minimal computational and data requirements, ultimately advancing the field together.

---

> > ### Author Response · Authors · 2025-08-04
> >
> > Dear Reviewer  LJos:
> >
> > We greatly appreciate the time and effort you dedicated to reviewing our paper. We have carefully addressed all your insightful suggestions and clarified any ambiguous points to improve our work. As the deadline for the discussion is nearing, could you kindly reconsider your evaluation based on the revised version? We are open to any further queries you might have and are eager to provide any additional information needed.
> >
> > Thank you for your understanding and support.
> >
> > Best regards,
> >
> > Authors

---

> > > ### Author Response · Authors · 2025-08-07
> > >
> > > Dear Reviewer LJos,
> > >
> > > We hope this message finds you well. We are writing to sincerely thank you for taking the time to review our rebuttal and for providing such insightful and constructive feedback on our submission. Your expertise and attention to detail are truly appreciated and have been incredibly beneficial in guiding our research.
> > >
> > > The suggestions you provided have opened up new perspectives and opportunities for enhancing the quality of our work. We are committed to implementing your recommendations to the best of our ability and are eager to explore the ideas you have mentioned further.
> > >
> > > Thank you once again for your invaluable contribution to our research. Your support and guidance are deeply appreciated, and we are hopeful that the revisions will meet your expectations.
> > >
> > > Sincerely,
> > >
> > > Author

---

> ### Author Response · Authors · 2025-08-08
>
> Dear Reviewer LJos,
>
> I hope this finds you well. As we near the end of the discussion period, we wanted to thank you once again for your thoughtful review. Please let us know if there are additional questions or aspects of the work you’d like us to address.
>
> We highly value your expertise and feedback, and we’re eager to continue discussing any unresolved points that could help refine and improve our research.
>
> Thank you for your time and for contributing to the review process.
>
> Sincerely,
>
> Author

---

### Author Response · Authors · 2025-08-09

We appreciate the reviewers’ insightful comments and constructive feedback on our manuscript. We are pleased to receive positive ratings from most of the reviewers. Furthermore, we are delighted to learn that the reviewers found the research problem to be significant and the core idea to be interesting (Reviewers LJos, TCBG, GJPB, and ULJw), the technical methodology to be efficient and impactful (Reviewers LJos, TCBG, and GJPB), and the experiments to be convincing and comprehensive (Reviewers TCBG, GJPB, and ULJw). Based on the reviews, we provide a general response to the points raised by multiple reviewers and individual responses below to address each reviewer’s concerns.

**(1)** Regarding the experimental design and supporting evidence, we have taken the following actions:
*   For Reviewers TCBG, LJos, and GJPB, we provided detailed ablation studies and highlighted results in the rebuttal (see *tables in the rebuttal, e.g., Tables 7 and 8*) to demonstrate the effectiveness of our **dynamic progressive alignment strategy**. This reduced computational costs to **40GB peak GPU memory** while achieving **SOTA performance** across omnimodal benchmarks.
*   For Reviewer TCBG, we clarified in the rebuttal (see *table in the rebuttal, e.g., Table 4*) the advantages of the **TGM-enhanced CTC decoder**, which balances real-time speech generation efficiency and competitive output quality compared to AR methods. This demonstrates that OpenOmni ensures both practical efficiency and high-quality synthesis.
*   For Reviewer LJos, we addressed the performance gaps for *text & symbols* and *count & quantity* tasks. We augmented the alignment training dataset with 50K targeted samples for each respective task. These targeted additions resulted in a significant boost (**33.33% to 50.89%**, see *table in the rebuttal*), validating the adaptability of OpenOmni to task-specific challenges.

**(2)** We have addressed the questions about the methodological details as follows:
*   For Reviewer GJPB, we clarified the novelty of our *dynamic progressive alignment strategy* and distinguished it from prior works (e.g., AnyGPT, Qwen-Omni). OpenOmni avoids the need for tri-modal data by using distinct two-modality alignment stages, delivering a **300% training efficiency improvement** compared to joint training.
*   For Reviewers TCBG, LJos, and GJPB, we explained how we mitigate *catastrophic forgetting*. Freezing the LLM during text-speech alignment and leveraging **WizardLM textual instructions** during image-text alignment ensured alignment objectives without significantly disrupting prior knowledge. This is validated in our rebuttal ablation studies (see *tables in the rebuttal, e.g., Tables 7 and 8*).
*   For Reviewer TCBG, we compared *CTC-DPO* with RLHF in quantitative experiments (see *rebuttal table*). Results showed that DPO is more stable, efficient, and equally effective. These findings confirm DPO’s suitability for **multi-turn emotional speech generation**, marking a significant advance over prior single-turn models like EMOVA.

**(3)** Missing or extended analysis:
*   For Reviewer LJos, we elaborated on the construction of *O2S-300K* and *EO2S-9K*, highlighting the **8,000 A100 GPU hours** used and unique advantages such as bilingual multi-turn training and emotional coherence. These datasets directly address limitations in prior works (e.g., VITA and LLaVA).
*   For Reviewer ULJw, we clarified the contributions of our work, emphasizing innovations such as the *TGM module*, the *MOE mechanism* for multilingual CTC training, and the *CTC-DPO algorithm* for emotional speech generation. These components create a scalable framework for open-source omnimodal research with minimal resource requirements.

**(4)** Novelty and contribution analysis:
We are delighted that multiple reviewers (e.g., TCBG, GJPB) recognized the novelty and value of our work. We summarize our key contributions below:
*   OpenOmni’s *dynamic progressive alignment strategy* addresses the computational challenges of tri-modal joint training. Using just **1.7M samples** and modest resources, OpenOmni delivers competitive results across omnimodal benchmarks, significantly lowering entry barriers for future research.
*   Our *TGM-enhanced CTC speech decoder* provides real-time speech generation while achieving competitive outputs comparable to AR methods (see **MOS evaluations and CER/WER results in the rebuttal**).
*   The *CTC-DPO algorithm* produces emotionally consistent, context-aware speech for multi-turn scenarios, avoiding the dependence on predefined emotional prompts used in prior systems. This advance is critical for naturalistic human-computer interactions.

We sincerely thank all the reviewers for their constructive suggestions and positive recognition of OpenOmni’s contributions. Please let us know if further explanations or details would be helpful!

Yours truly,
Authors of #8907

---

### Note · Authors · 2025-08-11

Dear Program Chairs (PC), Senior Area Chairs (SAC), Area Chairs (AC), and Reviewers,

We sincerely thank all the reviewers for their constructive feedback, which has refined our manuscript. We are encouraged by the positive evaluations and thoughtful comments, and deeply appreciate the valuable suggestions that clarified and improved key aspects of our work. This recognition highlights the significance and robustness of our approach.

The core contribution of our work lies in introducing an efficient, fully open-source omnimodal learning paradigm under data-scarce and resource-constrained conditions. OpenOmni achieves significant performance improvements over previous SOTA methods with fewer computational resources and without relying on tri-modal data, leveraging a dynamically progressive alignment strategy. Additionally, we designed a lightweight CTC speech decoder enhanced with TGM, supporting multi-turn dialog emotional consistency and multilingual generation. This decoder balances real-time decoding speed with competitive speech quality, significantly expanding practical applications and gaining adoption in subsequent works.

Compared to previous methods, our approach eliminates the need for tri-modal data joint training and requires only up to 40GB peak GPU memory. The progressive alignment method improves training efficiency by 300% while maintaining SOTA alignment performance, lowering the barrier for the open-source community to optimize omnimodal learning. Furthermore, OpenOmni's TGM-enhanced CTC module supports real-time multi-turn emotional speech generation and breaks single-turn limitations, enhancing user experience across diverse scenarios.

Beyond these innovations, we contribute 8K hours of high-quality bilingual multi-turn speech interaction data (O2S-300K) and multi-turn emotional consistency DPO training data (EO2S-9K) to support the open-source community, providing vital resources for speech generation and emotion alignment tasks.

Finally, we sincerely thank all reviewers for their invaluable feedback, which has greatly improved the rigor and clarity of our manuscript. If accepted, we will carefully incorporate all suggestions into the final version. We believe our work offers a novel and efficient paradigm for omnimodal learning, benefiting both the field and the open-source community.

Sincerely,
The Authors

---

### Decision · Program_Chairs · 2025-09-17

**Decision:**

Accept (poster)

**Comment:**

The paper introduces OpenOmni, an open-source omnimodal large language model (LLM) framework that integrates vision, text, and speech modalities, with a focus on real-time emotional speech synthesis and efficient multimodal alignment.

Key Contributions:

•	Modality-progression training: A two-stage alignment strategy enables effective tri-modal integration without requiring tri-modal data.

•	Real-time emotional speech synthesis: Achieved via a lightweight non-autoregressive decoder optimized with Direct Preference Optimization (DPO).

•	Efficiency and performance: Outperforms state-of-the-art models like VITA with 5× fewer training samples, smaller model size, and sub-second latency.

•	Benchmark results: Strong performance across OmniBench and other multimodal benchmarks, with improvements in emotion classification accuracy.

Reviewers’ overall attitudes:

•	Reviewers appreciated the technical depth, implementation clarity, and practical relevance of the work.

•	The rebuttal addressed key concerns, including architectural choices and training strategies.

•	While some noted that certain components build on prior work, the overall contribution is seen as valuable for the open-source community.

•	Reviewers generally raised their scores, with one moving to a borderline accept and others expressing strong support.